# Geometry-Aware visualization of high dimensional Symmetric Positive Definite matrices

**Thibault de Surrel**                                   *thibault.de-surrel@lamsade.dauphine.fr*
*LAMSADE, CNRS,*
*PSL Univ. Paris-Dauphine,*
*Paris, France*

**Sylvain Chevallier**                              *sylvain.chevallier@universite-paris-saclay.fr*
*LISN,*
*University Paris-Saclay,*
*Gif-sur-Yvette, France*

**Fabien Lotte**                                                    *fabien.lotte@inria.fr*
*Inria center at the University of Bordeaux, LaBRI,*
*Talence, France*

**Florian Yger**                                                  *florian.yger@insa-rouen.fr*
*LITIS*
*INSA Rouen-Normandy,*
*Rouen, France*

**Reviewed on OpenReview:** *https://openreview.net/forum?id=DYCSRf3vby*

## Abstract

Symmetric Positive Definite (SPD) matrices are pervasive in machine learning, from data features (such as covariance matrices) to optimization process. These matrices induce a Riemannian structure, where the curvature plays a critical role in the success of approaches based on those geometries. Yet, for ML practitioners wanting to visualize SPD matrices, the existing (flat) Euclidean approaches will hide the curvature of the manifold. To overcome this lack of expressivity in the existing algorithms, we introduce Riemannian versions of two state-of-the-art techniques, namely t-SNE and Multidimensional Scaling. Therefore, we are able to reduce a set of $c \times c$ SPD matrices into a set of $2 \times 2$ SPD matrices in order to capture the curvature information and avoid any distortion induced by flattening the representation in a Euclidean setup. Moreover, our approaches pave the way for targeting more general dimensionality reduction applications while preserving the geometry of the data. We performed experiments on controlled synthetic dataset to ensure that the low-dimensional representation preserves the geometric properties of both SPD Gaussian and geodesics. We also conduct experiments on various real datasets, such as video, anomaly detection, brain signal and others.

## 1 Introduction

Covariance matrices are being used in a lot of different scientific areas to better understand how multiple random variables are related to each other. Such matrices can be used in process control (Willjuice Iruthayarajan & Baskar, 2010), fault detection (Russell et al., 2000), biomedical image analysis (Pennec, 2020; Kobayashi et al., 2023) or Brain Computer Interfaces (BCI) (Lotte et al., 2018). Unfortunately, as soon as one manipulates a point cloud of covariance matrices each summarizing more than two random variables, it becomes impossible to visualize the whole point cloud in a 2 or 3 dimensional space. A common way

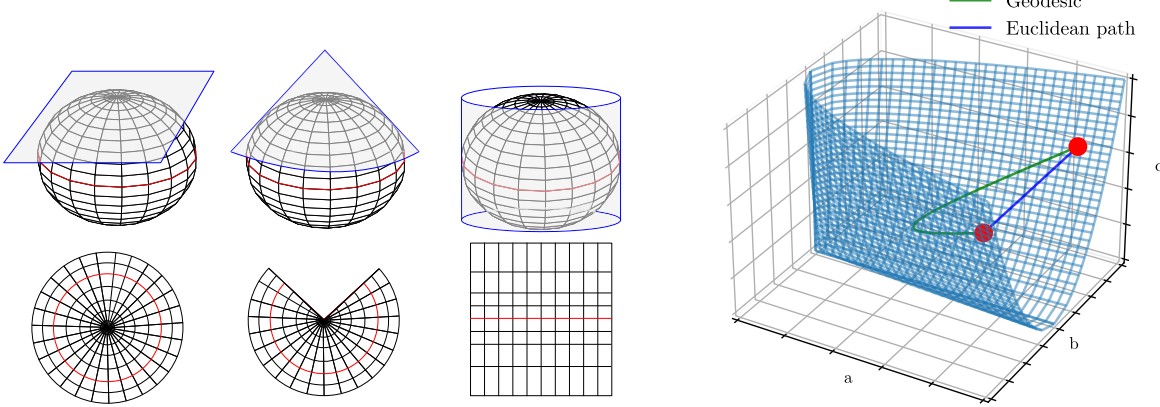

(a) Different examples of projections of the curved two-dimensional surface of a globe on a plane. All three of these projections distort in one way or another the surface.

(b) The set of $2 \times 2$ SPD matrices. The green path is the geodesic between two SPD matrices and the blue path is the usual Euclidean path.

Figure 1: Illustration of the distortion induced by a projection and the set of $2 \times 2$ SPD matrices.

to overcome this issue is to use a dimensionality reduction algorithm. The goal of such algorithms is to transform data from a high-dimensional space into a low dimensional space so that the low dimensional representation contains some meaningful properties of the original data. The ideal case is when we reduce sufficiently the dimension so that we are able to visualize the low-dimensional representation (for example, the low dimensional space is $\mathbb{R}^2$ or $\mathbb{R}^3$).

When dealing with covariance matrices, the natural geometry being used is the *affine invariant Riemannian geometry* (Bhatia, 2007; Pennec et al., 2006). Indeed, covariance matrices have a special structure, they are *symmetric*, *definite* and *positive* (SPD), therefore they live on the Riemannian manifold of SPD matrices. This manifold is not flat, contrary to usual Euclidean spaces (see Figure 1b to visualize the set of $2 \times 2$ SPD matrices and the geodesic between two points). This curvature implies that one can not reproduce accurately a point cloud sampled in a Riemannian space into a Euclidean one. For example, it is well known (Mulcahy & Clarke, 2001) that one can not project the Earth (the curved two-dimensional Riemannian surface of a sphere) on a plan without distorting the surface in some way and to some extent (see Figure 1a for some examples of projection). The same phenomenon appears when dealing with SPD matrices. A noticeable disadvantage of considering the Euclidean geometry when dealing with SPD matrices is the *swelling effect* (Arsigny et al., 2006). The swelling effect tells us that the determinant of the Euclidean mean can be strictly larger than the original determinants. As the determinant is an important measure of the dispersion of the associated multivariate random variable, we would like this dispersion to be correctly summarized in the mean. This negative effect disappears when considering the correct Riemannian geometry and its associated Riemannian mean.

Certain domains rely profoundly on covariance matrices such as BCI, where the goal is to translate brain signals into computer commands. In this context, the covariance matrices of ElectroEncephaloGraphy (EEG) brain signals are used as EEG representations to classify BCI users' mental commands. In BCI, it has been shown that using the Riemannian geometry is the correct way to deal with SPD matrices, and geometry aware classifiers have greatly improved the state of the art (Yger et al., 2017; Lotte et al., 2018; Barachant et al., 2013; Chowdhury & Andreu-Perez, 2021; Roy et al., 2022). However, for visualizing a set of SPD matrices coming from a BCI experiment, we may be tempted to use Euclidean tools such as the Euclidean t-SNE (Nishimoto et al., 2020; Zanini et al., 2018). Unfortunately, by doing so, the obtained visualizations do not take into account the Riemannian geometry that works so well for classification.

In this paper, we adapt two Euclidean algorithms: the *Multidimensional Scaling* (MDS) and the *t-SNE* to a Riemannian setting. We want to reduce a set of $c \times c$ SPD matrices while preserving the curvature of the original space into a lower dimension embedding, with $2 \times 2$ SPD matrices as an ideal candidate for visualization purpose. After adapting MDS and t-SNE to reduce matrices of the SPD manifold, we show

that the properties of the high-dimensional point cloud stay true in the small dimensional space of $2 \times 2$ SPD matrices. Specifically, we show that the important affine-invariance propriety is preserved and that geodesics and SPD Gaussians are correctly reproduced.

The paper is organized as follows: we start by reviewing the related works on dimensionality reduction in Section 2. Then, we introduce the Riemannian geometry on the set of SPD matrices in Section 3. The core of the paper is Section 4 where we describe the two Riemannian versions of the MDS and the t-SNE. We give some theoretical results on these algorithms in Section 5 and 6. In Section 7, we test our algorithms on synthetic datasets and real-life datasets from 4 different applications. We compare the proposed Riemannian versions of the algorithms against their Euclidean counterparts as well as against other geometry aware dimensionality reduction methods. Finally, we argue on our choice of the affine-invariant metric on the manifold of SPD matrices in section 8.

## 2    Related works

Dimension reduction (DR) is not a new topic in data analysis and machine learning. DR methods can be classified into two main categories: linear and non-linear methods.

**Linear methods:** The most famous linear method is probably the Principal Component Analysis (PCA) (Hastie et al., 2009) which seeks directions that best explain the original data. Another linear method is the Independent Component Analysis (ICA) (Hastie et al., 2009), that aims to separate a multivariate signal into additive, independent components. A supervised DR technique is Linear Discriminant Analysis (LDA) (Izenman, 2008) that seeks to find a linear combination of features that maximizes the separation between different classes in the data.

**Non-linear methods:** For non-linear methods, we have the Multidimensional Scaling (MDS) (Borg & Groenen, 1997) and t-SNE (van der Maaten & Hinton, 2008) that will be detailed in more depth in Section 4. Other non-linear methods include Kernel Principal Component Analysis (KPCA) (Schölkopf et al., 1997) which is an extension of traditional PCA that uses the kernel method to better work in high-dimensional spaces. Locally Linear Embedding (LLE) (Roweis & Saul, 2000) seeks to preserve local relationships within the high-dimensional data. It reconstructs each data point as a linear combination of its nearest neighbors. Another method whose goal is to find the intrinsic structures of data from a non-linear manifold is ISOMAP (Tenenbaum et al., 2000). It is a generalization of the MDS algorithm where the initial distances are computed using a neighborhood graph. A well known geometry-aware method is Uniform Manifold Approximation and Projection (UMAP) (McInnes et al., 2020) based on Riemannian geometry and tools from algebraic topology. The core idea is similar to t-SNE, that is to compute similarity matrices in the high and in the low-dimensional space and then optimize the low-dimensional one. The differences with t-SNE lie in the way the similarities are computed and the way they are optimized (UMAP uses the cross-entropy whereas t-SNE uses the Kullback-Leibler divergence). Finally, another method that focuses on discovering the underlying manifold of the data by modeling diffusion processes is Diffusion Maps (Coifman & Lafon, 2006). For most of these methods, a common assumption is that the data lies on an unknown manifold, and the goal is to learn this unknown manifold. In our case, instead, the data lies on a known manifold, the manifold of SPD matrices, and we leverage this information in our approaches.

**Riemannian methods:** Of course, some of these algorithms have been adapted to be used with data living on a known manifold by taking into account the geometry inherited by it. The PCA algorithm has been generalized to a manifold setting by Fletcher et al. (2004) to build the Principal Geodesic Analysis (PGA). Instead of seeking for the best directions to explain the data, PGA identifies the principal geodesics, which are the curves representing the most significant directions of variation on the manifold. Other algorithms have been extended to a Riemannian setting such as Locally Linear Embedding (LLE) (Saul & Roweis, 2003), and Hessian LLE (HLLE) (Goh & Vidal, 2008).

**Hyperbolic methods:** Some works have been done to reduce data living on hyperbolic spaces, as such spaces can be used in the representation learning of word embeddings (Nickel & Kiela, 2017) or visual inputs

(Khrulkov et al., 2020). For example, a PCA fitted for data lying on a hyperbolic space has been proposed by Chami et al. (2021) called HoroPCA. The difference with the PGA is that HoroPCA looks for directions rather than subspaces that explain the data. A similar work to ours is the adaptation of the t-SNE algorithm to hyperbolic spaces called CO-SNE by Guo et al. (2022).

**Methods for SPD matrices:**  Some algorithms have been especially designed for the manifold of SPD matrices. In Harandi et al. (2014), they learn a mapping from a high-dimensional manifold to a lower dimensional one without relying on tangent space approximations of the manifold. This algorithm demonstrated good results on image and video databases. Another extension of the PCA to the SPD manifold was proposed in Horev et al. (2017). This truly Riemannian PCA takes into account the curvature of the space and preserves the global properties of the manifold. The authors tested their algorithm on BCI data, as we will do in part 7.4. BCI is an interesting motivation to build DR algorithms for SPD matrices as the dimension of the covariance matrices that is dealt with grows quadratically with the number of sensors used to record the EEG. A lot of algorithms have been built with this application in mind (Coelho Rodrigues et al., 2017; Davoudi et al., 2017; Tanaka et al., 2016; Peng et al., 2023; Xie et al., 2017). Some work focused on the classification of SPD matrices (Huang & Van Gool, 2016; Brooks et al., 2019). It is possible to use SPDNet to perform dimensionality reduction, but it requires class information to learn the adequate low dimension representation, hence limiting scope of applications. Moreover, the learned embedding would be Euclidean therefore losing precious information about the geometry. The main difference between all the above algorithms and our approach is that their final goal is to classify the SPD matrices whereas our goal is to do visualization. Their goal is to preserve as much structure as possible, while increasing the separation between the classes to classify the data in the low-dimensional manifold better than in the original one. Our goal is different: we want to visualize the original dataset, with a 2D or 3D visualization reflecting the structure of the original space. In López et al. (2021), the authors develop a vector-valued distance and gyrocalculus for SPD matrices. The visualizations that they obtain are a by-product of the theory they develop. Moreover, and unlike our work, the visualizations they produce represent distances and angles of the high-dimensional SPD matrices. In our work, we reduce directly the SPD matrices in low-dimension and in a Riemannian manifold corresponding to the type of data we are working with (SPD matrices).

## 3   A primer on SPD matrices

**The Riemannian geometry of SPD matrices (Bhatia, 2007; Pennec et al., 2006)**
A $c \times c$ covariance matrix belongs to the set $\mathcal{P}_c$ of *symmetric*, *positive definite* (SPD) matrices:

$$\mathcal{P}_c = \left\{ X \in \mathbb{R}^{c \times c} | X = X^\top, \ \forall u \in \mathbb{R}^c \setminus \{0\}, \ u^\top X u > 0 \right\}$$

As $\mathcal{P}_c$ is an open and convex set of the set of symmetric $c \times c$ matrices $\mathbb{S}_c$, it is a *submanifold* of dimension $c(c+1)/2$. Moreover, its tangent spaces $T_X \mathcal{P}_c$ are identified with $\mathbb{S}_c$. Let us now define the following scalar product on each tangent space of $\mathcal{P}_c$:

**Definition 3.1.** Let $X \in \mathcal{P}_c$. We define the following *scalar product* on $T_X \mathcal{P}_c$:

$$\forall U, V \in T_X \mathcal{P}_c, \ \langle U, V \rangle_X = \operatorname{tr}(X^{-1} U X^{-1} V). \tag{1}$$

This metric is called the *Affine Invariant Riemannian Metric* (AIRM) as, if $A$ is an invertible matrix, $\langle AUA^\top, AVA^\top \rangle_{AXA^\top} = \langle U, V \rangle_X$. The application $X \mapsto \langle \cdot, \cdot \rangle_X$ being continuous, when $\mathcal{P}_c$ is endowed with this metric, it becomes a Riemannian manifold. Moreover, the manifold $\mathcal{P}_c$ is complete and therefore, each pair of points $(X, Y)$ can be connected by a unique minimizing geodesic:

**Proposition 3.2.** *Let $X, Y \in \mathcal{P}_c$. The expression of the unique geodesic $\Gamma$ linking them is:*

$$\forall t \in [0, 1], \ \Gamma(t) = X^{1/2} \left( X^{-1/2} Y X^{-1/2} \right)^t X^{1/2}.$$

This geodesic is the shortest path from $X$ to $Y$ on the manifold. The *Riemannian distance* between $X$ and $Y$, denoted $\delta$, is the length of the geodesic $\Gamma$ connecting $X$ to $Y$ (with log being the matrix logarithm):

$$\delta(X, Y) = \| \log(X^{-1/2} Y X^{-1/2}) \|. \tag{2}$$

An important property of the Riemannian distance defined above is its affine invariance property that inherits from the affine invariance of the metric:

**Proposition 3.3** (Affine invariance of the AIRM distance (Bhatia, 2007)). *Let $X, Y \in \mathcal{P}_c$ and $R$ be an invertible matrix of size $c \times c$, then one has $\delta(RXR^\top, RYR^\top) = \delta(X, Y)$.*

One can find more information on the Riemannian geometry of SPD matrices in appendix A. It should be noted that the AIRM geometry on the manifold of SPD matrices coincides (up to a factor $1/2$) with the Fisher-Rao metric on Multivariate Normal (Skovgaard, 1984). Another famous geometry used on the set $\mathcal{P}_c$ is the Log-Euclidean geometry (Arsigny et al., 2006). We argue on our choice of the AIRM metric in section 8.

**The special case of $2 \times 2$ SPD matrices**

Now that we have developed the general theory of $c \times c$ symmetric, positive definite matrices, let us focus on the special case of $\mathcal{P}_2$, the space of $2 \times 2$ SPD matrices. Indeed, it has a specific structure:

$$\mathcal{P}_2 = \left\{ \begin{pmatrix} a & b \\ b & c \end{pmatrix} \text{ s.t. } a > 0, \ c > 0, \ ac > b^2 \right\}.$$

Therefore, one can represent in $\mathbb{R}^3$ a $2 \times 2$ SPD matrix using $(a, b, c)$ and, it will lie strictly inside the cone of equation $b = \pm\sqrt{ac}$ where $a, c > 0$. We give a representation of this cone in Figure 1b.

# 4 Riemannian dimensionality reduction

In this section, we describe the two algorithms we studied and adapted to a Riemannian setting: the *Multidimensional Scaling* (MDS) and the *t-SNE*. The main difference between the two algorithms is that the MDS is a distance-based method whereas the t-SNE is a similarity-based method.

## 4.1 Riemannian Multidimensional Scaling

The goal of the *Multidimensional Scaling* (MDS) algorithm (Borg & Groenen, 1997) is to reproduce the pairwise distances between the high-dimensional point cloud and the low-dimensional one.

Given $N$ SPD matrices $X_1, ..., X_N$ of size $c \times c$, we can compute their distance matrix $D$ where $D_{ij} = \delta(X_i, X_j)$. We recall that $\delta$ is the Riemannian distance defined in Eq. 2. We are then looking for $N$ SPD matrices $Y_1, ..., Y_N$ of size $2 \times 2$ that minimize the stress function

$$S_R(Y_1, ..., Y_n) = \sum_{i<j} (\delta(Y_i, Y_j) - D_{ij})^2. \tag{3}$$

The goal is to find points in the low dimensional space $\mathcal{P}_2$ such that the pairwise distances are preserved as well as possible. It is a natural DR problem formulation, as we want the point cloud in low dimension $(Y_j)_j$ to "look" like the point cloud in high dimension $(X_i)_i$ as much as possible.

The stress function $S_R$ defined in Eq. 3 goes from $\mathcal{P}_2^N$ into $\mathbb{R}$ so, we can apply a Riemannian gradient descent algorithm (see Appendix B for more details). To be precise, we apply this Riemannian gradient descent algorithm on the product manifold $\mathcal{P}_2 \times \cdots \times \mathcal{P}_2$ which is also a manifold. For this algorithm, we need to compute the gradient of our cost function, here $S_R$. So, having in mind the gradient of the Riemannian distance as given in Proposition C.1, we have the following gradient:

$$\nabla_{Y_i} S_R(Y_1, ..., Y_N) = 2 \sum_{j \neq i} \left( \frac{D_{ij}}{\delta(Y_i, Y_j)} - 1 \right) \mathrm{Log}_{Y_i}(Y_j)$$

where we recall that $\mathrm{Log}_{Y_i}$ is the Riemannian logarithm as given in Proposition A.1. Details on the computation of the gradient can be found in Appendix C.2.

## 4.2 Riemannian t-SNE

We are now going to look into another DR algorithm this time based on a probabilistic framework. It was introduced in van der Maaten & Hinton (2008) and is called *t-SNE*. SNE stands for Stochastic Neighbor Embedding. The main difference with the MDS algorithm is that the MDS focuses on having a good low-dimensional representation of points that are far away. Indeed, in the cost function $S_R$ defined in Eq. 3, if two points $X_i$ and $X_j$ in the high-dimensional space are far apart, the quantity $\delta(X_i, X_j)$ will have a lot of importance in the cost function, way more than the distance of two points close to each other. On the contrary, we will see that the t-SNE algorithm focuses on retaining the local structure of the data.

Given a set of $N$ high-dimensional data points $X_1, ..., X_N$ in $\mathcal{P}_c$, we want to compute conditional probabilities that represent similarities. In the Euclidean setting, the similarity of $X_j$ to $X_i$ is the conditional probability $p_{j|i}$ that $X_i$ would pick $X_j$ as its neighbor if they were picked in proportion to their probability density under a Gaussian centered at $X_i$. To generalize this definition to SPD matrices, we use the generalization of the Gaussian to the manifold of SPD matrices introduced in Said et al. (2016) (more details in Appendix A.2 on this distribution). Thus, $p_{j|i}$ is given by:

$$p_{j|i} = \frac{\exp(-\delta(X_i, X_j)^2/2\sigma_i^2)}{\sum_{k \neq i} \exp(-\delta(X_i, X_k)^2/2\sigma_i^2)} \tag{4}$$

where $\sigma_i^2$ is the variance of the Gaussian centered at $x_i$. To choose $\sigma_i$, we perform a binary search that produces a $P_i = (p_{j|i})_{j=1,...,N}$ with a fixed perplexity chosen by the user. The perplexity is a parameter defined the same way as in the original t-SNE (van der Maaten & Hinton, 2008). It is defined as $\text{Perp}(P_i) = 2^{H(P_i)}$ where $H(P_i)$ is the Shannon entropy of $P_i$ (see Shannon (1948) for its definition). We then symmetrize the conditional probabilities by setting $p_{ij} = \frac{p_{j|i}+p_{i|j}}{2N}$ when $i \neq j$ and $p_{ii} = 0$.

Then, the t-SNE algorithm aims at learning $Y_1, ..., Y_n \in \mathcal{P}_2$, that reflect the similarities $p_{ij}$ as well as possible. The Euclidean setting uses a Student t-distribution with one degree of freedom as the heavy-tailed distribution in the low-dimensional map. This way, dissimilar data points in high dimension will be modeled far away in low dimension. We will extend this distribution to the manifold of SPD matrices and define the joint probabilities $q_{ij}$ as:

$$q_{ij} = \frac{\left(1 + \delta(Y_i, Y_j)^2\right)^{-1}}{\sum_{k \neq l}(1 + \delta(Y_k, Y_l)^2)^{-1}}. \tag{5}$$

We then want to minimize the mismatch between the two conditional probabilities $p_{ij}$ and $q_{ij}$. To measure this, the Riemannian t-SNE (as the Euclidean t-SNE) uses the Kullback-Leibler (KL) divergence and tries to minimize it. The cost function $C_R$ is then:

$$C_R = \sum_{i=1}^{N} \text{KL}(P_i||Q_i) = \sum_{i=1}^{N} \sum_{j \neq i} p_{ij} \log \frac{p_{ij}}{q_{ij}}. \tag{6}$$

As for the MDS, a Riemannian gradient descent method is performed in order to minimize the KL divergence with respect to the points $Y_i$. The gradient of $C_R$ with respect to $Y_i$ is given by:

$$\nabla_{Y_i} C_R = -4 \sum_{j=1}^{N} \frac{p_{ij} - q_{ij}}{1 + \delta(Y_i, Y_j)^2} \text{Log}_{Y_i}(Y_j).$$

More details on this derivation are given in Appendix C.3.

## 5 Affine invariance of the algorithms

In this section, we give a theoretical result on the two algorithms we proposed in Section 4. This result regards the affine invariance of our problem and of our algorithms. Indeed, we recall the important property of the affine invariance of the AIRM distance 3.3. Therefore, as both the MDS and the t-SNE depend only

on the distances between the point in high dimension, the two algorithms are also affine-invariant. We detail only the Riemannian MDS case, the results and proof being the same for the Riemannian t-SNE.

**Proposition 5.1** (Affine invariance of the algorithms). *Let $X_1, ... X_N$ be points in $\mathcal{P}_c$ and $R$ be an invertible matrix.*

1. *Let $(\tilde{X}_1, ..., \tilde{X}_N) = (RX_1 R^\top, ..., RX_N R^\top)$. Then, the set of solutions of the Riemannian MDS when the initial high-dimensional points are $(X_1, ..., X_N)$ is the same as the set of solutions of the Riemannian MDS when the initial high-dimensional points are $(\tilde{X}_1, ..., \tilde{X}_N)$.*

2. *Let $(Y_1, ..., Y_N)$ be a solution of the Riemannian MDS when the initial high-dimensional points are $(X_1, ..., X_N)$. Then $(RY_1 R^\top, ..., RY_N R^\top)$ is also a solution of the problem.*

The proof can be found in Appendix D. This affine invariance property is particularly useful for EEG analysis. Indeed, full rank spatial filters are often used to process EEG and are affine transformations (Congedo et al., 2017). Thus, comparing covariance matrices in the sensor space (i.e., obtained without applying spatial filter) amounts to directly comparing covariance matrices in the source space (i.e, that would be obtained with a spatial filter, e.g., with inverse solutions). Moreover, when comparing the results of a same subject over two sessions, it is likely that the EEG electrodes are not exactly at the exact same position on the head. Therefore, there exists an affine transformation between the covariance matrices recorded over two sessions. As our algorithms are affine invariant, the reduced point clouds will still be comparable between the two sessions and will be representative of the actual brain activity, no matter the filter applied. Other Euclidean DR algorithms do not have this property of affine invariance, therefore, in the situations described above, some artifacts could appear when comparing two point clouds different up to an affine transformation.

## 6 A convergence result

We want to comment on the convergence of the Riemannian MDS. For this, let us consider $Y \in \mathcal{P}_c$, $d \geq 0$ and let us define the following sub function $f_{Y,d}$ of the Riemannian MDS:

$$f_{Y,d}(X) = (\delta(X, Y) - d)^2 .$$

As one can see using Eq. 3, the Riemannian MDS loss can be rewritten as the sum of such functions. Unfortunately, this function is not g-convex (see Appendix B for the definition of a g-convex function and a counter-example for $f_{Y,d}$), so we do not have the usual convergence results. However, the function $f_{Y,d}$ can be rewritten the following way:

$$f_{Y,d}(X) = \underbrace{\delta(X, Y)^2 + d^2}_{g(X)} - \underbrace{2d\delta(X, Y)}_{h(X)} . \tag{7}$$

Moreover, we know from Sra & Hosseini (2015) (Example 20) that $X \mapsto \delta(X, Y)$ is g-convex. Therefore, as $d \geq 0$, both $g$ and $h$ are convex and the function $f_{Y,d}$ can be written as a difference of g-convex functions. Moreover, $g$ and $h$ are also lower semi-continuous and proper functions. One can then use the Difference of Convex Algorithm (DCA) proposed in Bergmann et al. (2023) to solve the sub-problem. In this case, the sequence of points generated by the DCA will converge to a critical point of the cost function $f_{Y,d}$.

## 7 Synthetic experiments

Now that we have described the two algorithms that we introduced, we can test them. We now show some experiments, first on synthetic datasets and then on real data. We used pyRiemann (Barachant et al., 2023) to deal with SPD matrices in Python[1].

---

[1]Our code is available at `https://github.com/thibaultdesurrel/riemannien_dimension_reduction`

### 7.1 The contestants

In these experiments, we assess how the algorithms we proposed in the previous section perform against other DR algorithms. We first want to confirm that using Riemannian geometry is better when dealing with SPD matrices. For this, we compare 3 versions of the MDS and the t-SNE:

- A Riemannian version: the input and output distances are computed using the AIRM.

- A Riemannian-Euclidean version: the input distances are computed using the AIRM, but the output are points from $\mathbb{R}^3$ equipped with the Euclidean structure.

- A Euclidean version: the input distances are computed using the Euclidean structure of symmetric matrices (the Frobenius norm) and the output are also Euclidean points in $\mathbb{R}^3$.

We also compare our algorithms with other DR algorithms. To do this, we compare the proposed algorithms with the Principal Geodesic Analysis (PGA) (Fletcher et al., 2004), UMAP (McInnes et al., 2020) and the Riemannian PCA (Horev et al., 2017). They are all described in Sec. 2. The output spaces of PGA and UMAP is the Euclidean space $\mathbb{R}^3$ while for the Riemannian PCA, it is $\mathcal{P}_2$ equipped with the AIRM. The input of the UMAP algorithm is the Riemannian distance matrix.

### 7.2 The evaluation metrics

As our goal is to build a visualization tool for datasets of SPD matrices, a first criterion is a qualitative one: how it looks. We will also use a quantitative metric: the trustworthiness coefficient (Venna & Kaski, 2001). It ranges from 0 to 1 (with 1 being the highest score) and is computed by ranking the order of each point's neighbors and comparing these ranks between the original high-dimensional space and the reduced low-dimensional space. Any nearest neighbors in the reduced space that differ from the original space are penalized according to their ranking in the input space. The number of nearest neighbors $k$ is a hyperparameter that will help us understand to what extent the local and global structures of the point cloud are preserved. One needs $k < N/2$ where $N$ is the total number of points.

### 7.3 Synthetic experiments

**Reducing SPD Gaussians**  For this first experiment, we reduce a mixture of three Gaussians on the set of SPD matrices. We sample $N = 225$ points from a mixture $G(\bar{X}_1, \sigma_1) + G(\bar{X}_2, \sigma_2) + G(\bar{X}_3, \sigma_3)$ where the means $(X_i)_{i=1,2,3}$ are the means of the 3 classes of the dataset AirQuality that is described in Section 7.4 and where $\sigma_1 = 1/2$, $\sigma_2 = 1/3$, $\sigma_3 = 1/4$. The points are balanced between the 3 Gaussians. We then use the different algorithms detailed in Sec. 7.1 to reduce this high-dimensional point cloud. The results[1] are given in Figure 2. We can see that the fully Riemannian algorithms place the different Gaussians along a geodesic (see Figure 1b to see what a geodesic looks like in the space of SPD matrices). Therefore, the blue and green Gaussians are further apart from each other than they are in a fully Euclidean setting. Between the two Riemannian versions, the Riemannian t-SNE produces a nicer visualization, the Gaussians being separated and less clustered at the origin. Finally, while the Riemannian-Euclidean algorithms are correctly separating the Gaussians, the resulting point cloud is in the Euclidean space $\mathbb{R}^3$, and not along geodesics as they should be. Thus, it does not bring any insight on how the Riemannian geometry affects the data.

**Reducing geodesics**  In this second experiment, we investigate how a high-dimensional geodesic $\Gamma$ in $\mathcal{P}_c$ is reduced using our algorithms. We want to see how close the reduced trajectory is to the corresponding geodesic in $\mathcal{P}_2$. More details on the setup are given in appendix E. In this experiment, we can only compare the Riemannian t-SNE with the Riemannian MDS as we need the reduced points to lie in $\mathcal{P}_2$. We conducted some experiments varying some parameters: the dimension $c$ of the high-dimensional geodesic, the noise $\varepsilon$ on its sampling and the number of points $N$ sampled.

---

[1]Those figures should be taken with caution, as they are only 2D representations of 3D point clouds. The interested reader can find some interactive plots in the supplementary materials.

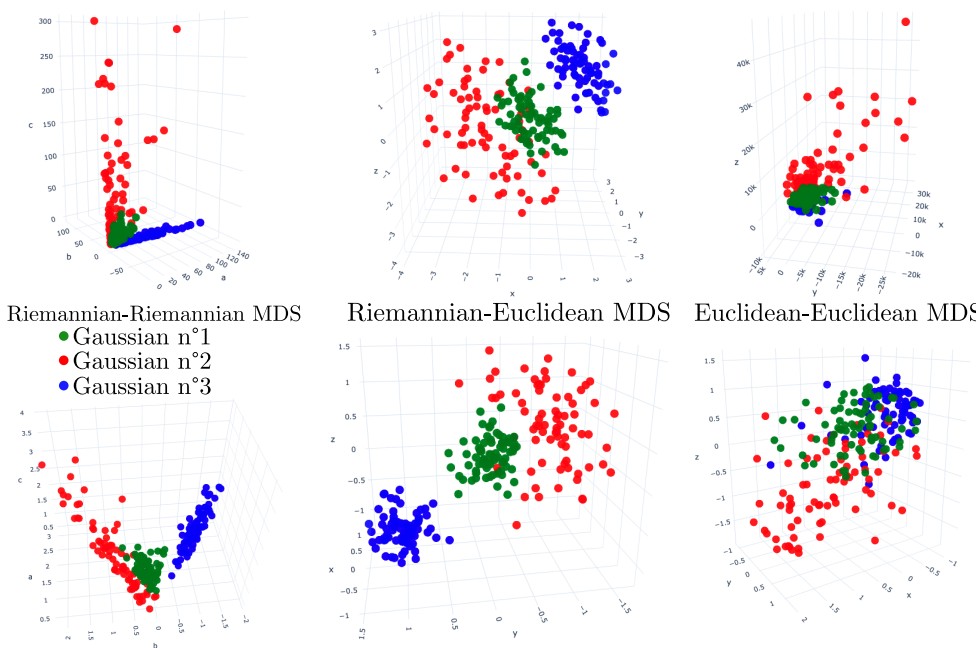

Figure 2: Results of the first synthetic experiment[1], a comparison of the reduction of a mixture of 3 Gaussians.

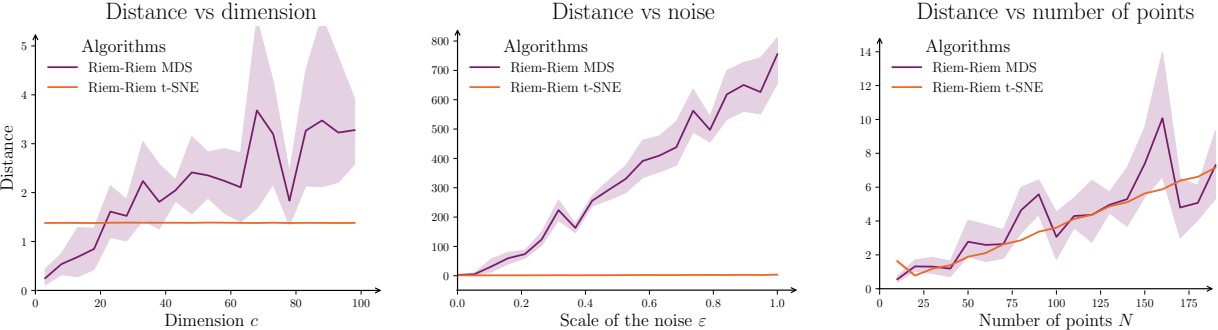

Figure 3: Results of the second synthetic experiment: comparison of the distances of the reduced geodesic by the t-SNE or the MDS algorithm to the real geodesic.

From a qualitative point of view, and based on the visual example given in Figure 11, both algorithms preserve the structure of the high-dimensional geodesic. However, the MDS seams to do a better job than the t-SNE. Indeed, at each extremity of the geodesic reduced using t-SNE, the points are less precisely on the geodesic whereas the points in the middle are exactly on the geodesic. This can be explained by the fact that the t-SNE focuses more on reproducing correctly points that are close to one another while the MDS gives an equal importance to each point. We want to compare the reduced trajectory to a true geodesic in $\mathcal{P}_2$. For this, we compute the geodesic going from one extremity to the other of the reduced trajectory. Then, we compare the distance between this true geodesic and the reduced trajectory. The results over 5 simulations are given in Figure 3. We can see that the Riemannian t-SNE is more robust than the Riemannian MDS. In fact, when varying the dimension of the high-dimensional space or the noise, the distance to the true geodesic remains virtually unchanged. However, as the number of points rises, the distance grows. This is coherent as the optimization problem becomes harder the more points there are. The MDS, on the other hand, is very noisy and the quality of the reduction worsens as the dimension or the noise grows. We must clarify that the t-SNE is not entirely robust to noise and changes in dimensionality. However, it demonstrates significantly

| Name | Domain | Number of matrices | Dimension | Ref. |
|---|---|---|---|---|
| BNCI2014001 | Brain Computer Interfaces | $288 \times 9$ subjects | $22 \times 22$ | (Tangermann et al., 2012) |
| BNCI2014002 | Brain Computer Interfaces | $160 \times 12$ subjects | $15 \times 15$ | (Steyrl et al., 2015) |
| AlexMI | Brain Computer Interfaces | $60 \times 8$ subjects | $16 \times 16$ | (Barachant, 2012) |
| AirQuality | Atmospheric pollutants | 102 | $6 \times 6$ | (Hua et al., 2021; Smith et al., 2022a) |
| FPHA | Video sequences of hand actions | 108 | $63 \times 63$ | (Garcia-Hernando et al., 2018; Wang et al., 2023) |
| TEP | Anomaly detection | 420 | $52 \times 52$ | (Downs & Vogel, 1993; Smith et al., 2022b) |

Table 1: Summary of the datasets used for the experiments.

greater robustness compared to MDS, which is highly sensitive to these two factors. The relative robustness of t-SNE to noise comes from the fundamental design of the algorithm as explained above.

### 7.4 Experiments on real datasets

**Setup** After testing our algorithms on synthetic datasets, we now study their behavior on 6 real datasets from different applications that are summarized in Table 1. More details on the datasets as well as on the setup are given in Appendix G. We used the different algorithms presented in Sec. 7.1 to reduce the datasets. For the Riemannian t-SNE (AIRM and Log-Euclidean), we chose a perplexity of $\frac{3}{4}N$ where $N$ is the total number of points. For the Euclidean t-SNE, we chose a fixed perplexity of 30 as it is the default parameter for the t-SNE of scikit-learn (Pedregosa et al., 2011) (see Appendix F for more insights on these choices).

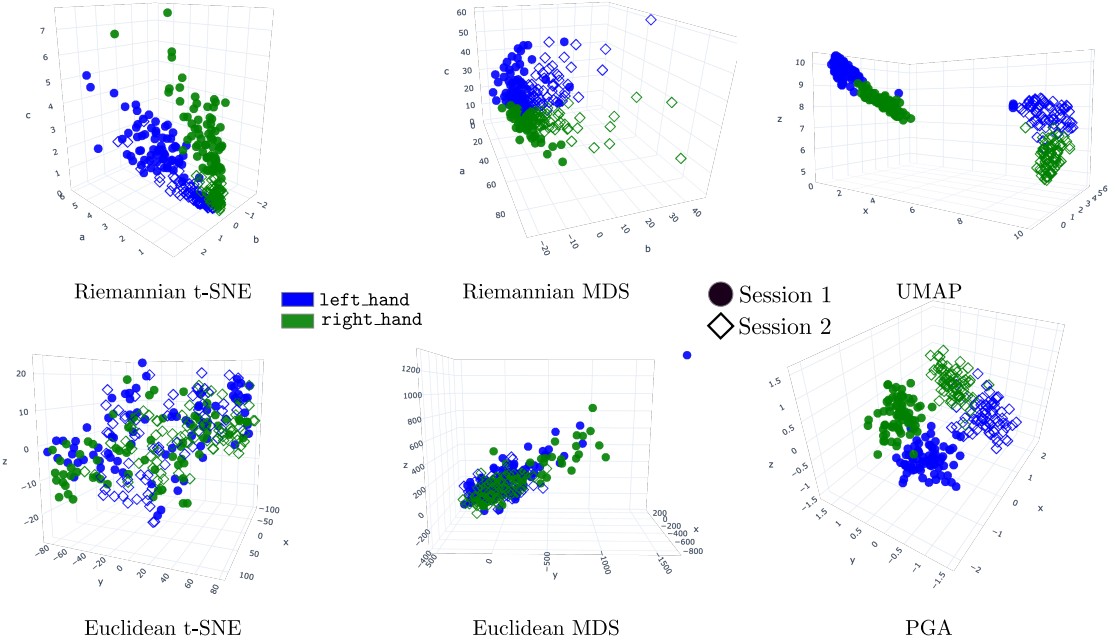

Figure 4: Result of the different algorithms on subject 8 of BNCI2014001 (Tangermann et al., 2012).

**Results** We give in Figure 4 the results of the different algorithms studied when reducing the subject 8 of the dataset BNCI2014001 (Tangermann et al., 2012) from a BCI experiment. The colors correspond to the classes and the shape of the markers corresponds to the two sessions during which this dataset was recorded. We see that the geometry aware algorithms (Riemannian t-SNE, Riemannian MDS, PGA and UMAP) are able to separate the two classes. We also clearly see that the two sessions are well separated. This is not surprising as the signals recorded by an EEG during a BCI experiment are sensitive to the user's mental state (Saha & Baumert, 2020; Krumpe et al., 2017), so it is likely that the data has shifted from one session to another. We also give in Figure 13 the results of the different algorithms on the dataset AirQuality.

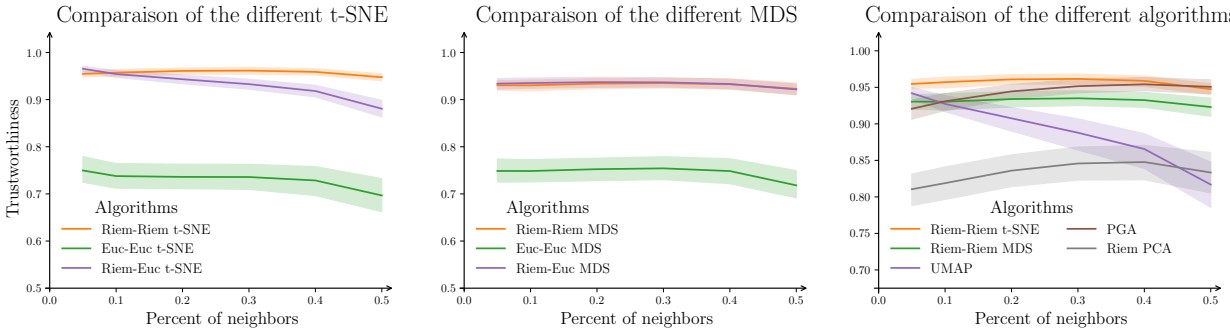

Figure 5: The results of the experiments on the real datasets: we plot the average of the trustworthiness coefficients over 6 datasets. The x-axis represents the size of neighborhood used to compute the trustworthiness coefficient as % of the total number of points.

| Dataset | Riem-Riem t-SNE | Euc-Euc t-SNE | Riem-Riem MDS | Euc-Euc MDS |
|---|---|---|---|---|
| BNCI2014001 | 198.34 ($\pm$101.58) | 274.92 ($\pm$6.99) | 733.20 ($\pm$598.40) | **19.08** ($\pm$35.21) |
| BNCI2014002 | 62.11 ($\pm$49.49) | 113.38 ($\pm$31.79) | 330.17 ($\pm$246.47) | **3.49** ($\pm$1.83) |
| AlexMI | 7.45 ($\pm$3.39) | 0.71 ($\pm$0.42) | 50.33 ($\pm$19.12) | **0.42** ($\pm$0.24) |
| AirQuality | 2.53 | 0.75 | 379.26 | **0.71** |
| FPHA | 10.58 | 2.31 | 383.45 | **0.81** |

Table 2: Computation time of the different algorithms on the different datasets. The time is given in seconds. For the BCI datasets, we displayed the mean time over all the subjects.

Our goal is to quantitatively see how the local and global structures of the high-dimensional point cloud are preserved. For this, we look at how the trustworthiness coefficient evolves when the neighborhood expands (sec. 7.2). We give in Figure 5 the results of this experiment and in Appendix H the detailed results. A first important observation is that considering the Riemannian geometry of SPD matrices is important as both the t-SNE and the MDS work better when the distances are computed using the AIRM distance of Eq. 2. When focusing on the different t-SNE, we see that the local structure (less than 10% of all the neighbors) is well-preserved using the Riemannian-Euclidean t-SNE but, as soon as the neighborhood gets bigger, the fully Riemannian t-SNE conserves better the structure. This is explainable as, locally, a Riemannian manifold can be approximated by a Euclidean space (the tangent space) but globally, the curvature induces some deformations of the space. This is exactly what we observe and therefore, the Riemannian t-SNE should be used when trying to reduce a dataset of SPD matrices. For the MDS, we were not able to see any quantitative differences between the fully Riemannian MDS and the Riemannian-Euclidean one.

When comparing our algorithms with PGA, UMAP and the Riemannian PCA, we clearly see that the Riemannian t-SNE is the best algorithm on these datasets. As UMAP does not have any *a priori* information on the geometry of the point cloud, it only manages to reproduce correctly the local structure of the point cloud and collapses as the neighborhood grows. PGA seems to be able to preserve the global structure of the point cloud, but is not very good when dealing with local structures. The Riemannian PCA greatly under-performs compared to the proposed algorithms. The Riemannian t-SNE works better than the Riemannian MDS, as we had already witnessed in the previous experiments. All in all, the Riemannian t-SNE is the best trade-off between preserving the local and the global structures of a set of high-dimensional SPD matrices.

## 7.5 Details on computation times

The computations were made on a Macbook Pro M3 with 36 Go of memory. The time taken by the different algorithms to reduce the various datasets are detailed in Table 2. The Euclidean algorithms are, of course, faster than the Riemannian ones as each iteration is longer to compute. Indeed, at each iteration of the gradient descent, the distance matrix between all the points needs to be computed. As the AIRM distance

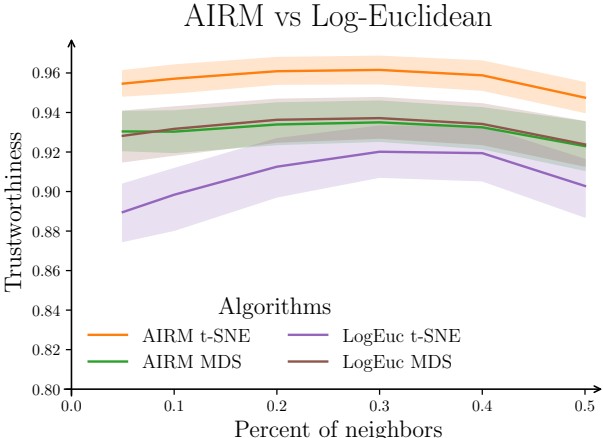

Figure 6: Comparing the two distances on the manifold of SPD matrices. The trustworthiness coefficient over the 6 datasets is averaged.

requires computing matrix inverse square roots and a matrix logarithm, it takes more time to compute than its Euclidean counterpart. However, one should remark that the dimension of the high-dimensional points has very little influence on the time each iteration takes to compute: at each iteration, only $2 \times 2$ SPD matrices are at stake, so the distance matrix is always computed on $2 \times 2$ matrices. We also remarked that the Riemannian t-SNE needed much less time to converge than the Riemannian MDS.

## 8    Affine invariant metric or Log-Euclidean metric ?

In section 3, we presented the *Affine Invariant Riemannian metric* (AIRM) for the manifold of SPD matrices. It is not the only metric that exists on the manifold $\mathcal{P}_c$. Another widely used (Huang et al., 2015; Chevallier et al., 2021) distance is the *Log-Euclidean* distance (Pennec et al., 2006) that preserves theoretical properties while requiring fewer computations. The main idea of the Log-Euclidean is to apply the standard Euclidean metric to the matrices after transforming them with the matrix logarithm. This transformation maps them to the tangent space at the identity matrix, which is a Euclidean space. We define this metric below.

**Definition 8.1** (Log-Euclidean distance)**.** The Log-Euclidean distance $\delta_{LE}$ between $X \in \mathcal{P}_c$ and $Y \in \mathcal{P}_c$ is:

$$\delta_{LE}(X, Y) = \| \log(X) - \log(Y) \|_F$$

In this section, we want to compare the Riemannian t-SNE and Riemannian MDS based on the AIRM metric (as done so far) and based on the Log-Euclidean distance described above. For this, we implemented versions of the t-SNE and the MDS that uses the Log-Euclidean distance instead of the AIRM distance. We refer to the two versions of the t-SNE as *AIRM t-SNE* and *LogEuc t-SNE* and likewise for the MDS. As the Log-Euclidean does not have the affine invariance property contrary to the AIRM distance, the built algorithms using the Log-Euclidean are not affine invariant.

We conducted the same experiments as in section 7.4 on the real datasets, and we show the results in Figure 6. For the t-SNE, it is clear that using the AIRM distance leads to better results than using the Log-Euclidean distance. For the MDS, changing the distance used does not lead to significant changes in the results. As we loose some important properties (such as the affine invariance) when computing distances with the Log-Euclidean distance without any result improvement, we conclude that one should use the AIRM distance when reducing SPD from a high-dimensional space to a low-dimensional one.

## 9 Conclusion

In this paper, we introduced a variant for t-SNE and MDS to handle data on the Riemannian manifold of SPD matrices. We showed that the Riemannian algorithms are able to retain important properties of the geometry such as the affine invariance. Moreover, we showed that we could cast our problem into a class of optimization problems that can be written as a difference of g-convex functions. In the experiments, we did not implement this DCA, but rather a Riemannian Gradient Descent (RGD) algorithm. We do not have any theoretical guarantees on the convergence of this problem using a RGD, however, in practice, we stopped the RGD when the gradient was small enough ($< 10^{-6}$). There are also no guarantees that the critical point reached is indeed a minimizer of our cost function. Finally, we tested our algorithms on synthetic and real-world data and demonstrated their usefulness. The Riemannian t-SNE is the most interesting in real-world applications. Hence, if the goal is to visualize a set of SPD matrices, and therefore we care about the faithfulness of the representation, our Riemannian t-SNE should be used. Furthermore, even if using a less computationally heavy distance on the manifold of SPD matrices (such as the log-Euclidean one), we showed that the best suited metric for our work is the AIRM one.

In future works, we consider several promising extensions and applications of our methods. First, the optimization algorithms are computationally expensive and may limit the use of our approaches for very large datasets. A stochastic optimization setting on manifolds (Bonnabel, 2013) is a promising solution. Our Riemannian dimensionality reduction could be used as a visual feedback tool to help users train controlling BCIs. This could help to tackle the issue of BCI illiteracy/deficiency (Allison & Neuper, 2010), in the spirit of Duan et al. (2021) but conforming with the AIRM geometry. Finally, our approaches focused on the case of SPD matrices, but it could be extended to other manifolds. For this, one would need a manifold that can be viewed and plotted in two or three dimensions such as Stiefel manifolds that can be used to visualize sets of subspaces (Yamamoto et al., 2021).

### Acknowledgments

This work was funded by the French National Research Agency for project PROTEUS (grant ANR-22-CE33-0015-01). Part of this work was carried out while Florian Yger was a member of PSL-Dauphine University, and he acknowledges the support of the ANR as part of the "Investissements d'avenir" program, reference ANR-19-P3IA-0001 (PRAIRIE 3IA Institute). Sylvain Chevallier is supported by DATAIA (ANR-17-CONV-0003)

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

# A    More on the Riemannian geometry of SPD matrices

In this section, we introduce some useful tools to study the Riemannian geometry of Symmetric Positive Definite (SPD) matrices. This section completes what was said in section 3.

## A.1    The exponential and logarithm mapping

The mapping Exp allows going from the tangent space $T_X \mathcal{P}_c$ to the manifold $\mathcal{P}_c$ and its inverse map Log are useful tools in Riemannian geometry. In the case of the SPD manifold $\mathcal{P}_c$, equipped with the affine invariant metric given at Eq. 1, the expression of the Exp and Log mappings are:

**Proposition A.1.** *Given $X, Y \in \mathcal{P}_c$ and $U \in T_X \mathcal{P}_c$, one has:*

$$\mathrm{Exp}_X(U) = X^{1/2} \exp\left(X^{-1/2} U X^{-1/2}\right) X^{1/2}$$
$$\mathrm{Log}_X(Y) = X^{1/2} \log\left(X^{-1/2} Y X^{-1/2}\right) X^{1/2}$$

(8)

### A.2 Distributions on the manifold of SPD matrices

In the sequel and in order to define the conditional probabilities for the Riemannian t-SNE, we need a Gaussian distribution on the space of symmetric positive definite matrices $\mathcal{P}_c$. A generalization of the usual Euclidean Gaussian distribution on $\mathcal{P}_c$ was introduced in Said et al. (2016). Let us consider $\bar{Y} \in \mathcal{P}_c$ and $\sigma > 0$.

**Definition A.2.** The *Gaussian distribution* $G(\bar{Y}, \sigma)$ on $\mathcal{P}_c$ is defined using its probability density function $p_{\bar{Y},\sigma}$ and $\zeta(\sigma)$ its *normalizing factor* as:

$$p_{\bar{Y},\sigma}(Y) = \frac{1}{\zeta(\sigma)} \exp\left[-\frac{\delta(Y,\bar{Y})^2}{2\sigma^2}\right].$$

The interpretation of the two parameters $\bar{Y}$ and $\sigma$ are the same as in the Euclidean case: $\bar{Y}$ is the center of mass of the distribution and $\sigma$ is a parameter that controls the dispersion of the distribution. An important property is that the normalizing factor $\zeta(\sigma)$ does not depend on the parameter $\bar{Y}$, only on $\sigma$. An expression of $\zeta(\sigma)$ can be found in proposition 4 of Said et al. (2016).

## B Optimizing on a Riemannian manifold

We introduce here basic tools for optimizing on a Riemannian manifold $\mathcal{M}$. More details on the following tools can be found in Boumal (2023).

### B.1 Riemannian Gradient Descent (RGD)

We study the following problem:

$$\min_{x \in \mathcal{M}} f(x)$$

where $f: \mathcal{M} \to \mathbb{R}$ is a smooth function called the *cost function*. We are going to take inspiration from the Euclidean gradient descent $x_{k+1} = x_k - \alpha_k \operatorname{grad} f(x_k) \quad k = 0, 1, 2....$ However, we need to adapt this algorithm to the Riemannian manifold $\mathcal{M}$. As $\mathcal{M}$ is not always a vector space, it is not guaranteed that, for a given $x_k$, $x_k - \alpha_k \operatorname{grad} f(x_k)$ still lies on the manifold. We need a way to project back this point onto the manifold. This is done using a *retraction*, an operator $R_x: T_x\mathcal{M} \to \mathcal{M}$ which projects a point from a given tangent space back onto the manifold. We can then use this retraction to adapt the Euclidean gradient descent:

**Algorithm B.1** (Riemannian gradient descent)**.** *Given a Riemannian manifold $\mathcal{M}$, an initial point $x_0 \in \mathcal{M}$ and a retraction $R$ on $\mathcal{M}$, we iterate*

$$x_{k+1} = R_{x_k}(-\alpha_k \nabla f(x_k)), \qquad k \in \mathbb{N}$$

*where, at each iteration $k$, we pick a step-size $\alpha_k$.*

To choose the step-size $\alpha_k$, the most common method is to use *line-search* where the goal is to minimize the function $g(t) = f(R_{x_k}(-t\nabla f(x_k)))$ approximately.

### B.2 Geodesic convexity

In a usual Euclidean setting, when one wants to have some results on an optimization problem, the first reflex is to check if the loss function is convex. As we are in a Riemannian setting, we need to adapt the notion of convexity: we introduce the notion of geodesically convex (or g-convex) function:

**Definition B.2.** A function $f: \Delta \to \mathbb{R}$ is *geodesically convex* (or g-convex) if $\Delta$ is geodesically convex subset of a manifold $\mathcal{M}$ and $f \circ \Gamma: [0,1] \to \mathbb{R}$ is convex for each geodesic segment $\Gamma: [0,1] \to \mathcal{M}$ whose image is in $\Delta$ (with $\Gamma(0) \neq \Gamma(1)$). This means that

$$\forall t \in [0,1], \ f(\Gamma(t)) \leq (1-t)f(\Gamma(0)) + tf(\Gamma(1)). \tag{9}$$

In our case, as the set $\mathcal{P}_c$ of symmetric positive definite matrices is convex and complete, it is geodesically convex and therefore, we can study the g-convexity of the loss functions we encounter.

## B.3  Counter example on the geodesically convexity of the sub-problem of the Riemannian MDS

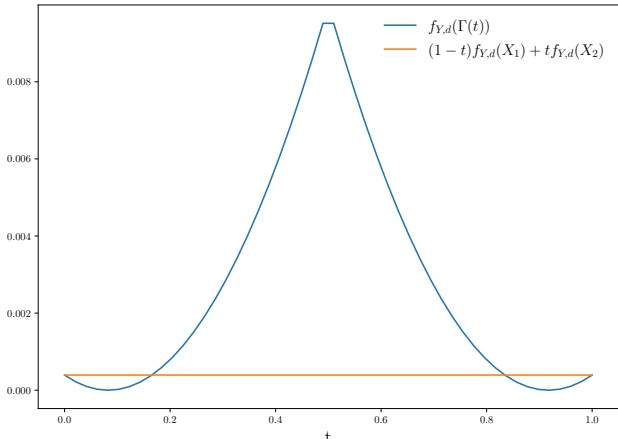

Figure 7: Counter example of the g-convexity of the sub-function $f_{Y,d}$ of the Riemannian MDS.

In this section, we prove the result stated in Section 6: the sub-function $f_{Y,d}$ of the Riemannian MDS is not g-convex. We recall that this sub-function is defined as:

$$f_{Y,d} = (\delta(X, Y) - d)^2.$$

with $Y \in \mathcal{P}_c$ and $d > 0$. For $f_{Y,d}$ to be g-convex, it needs to satisfy the inequality 9 for all geodesic $\Gamma$. However, if we consider a geodesic that passes through $Y$, this inequality is not satisfied as shown in Figure 7. For this plot, we chose two random SPD matrices $X_1$ and $X_2$ and chose $Y$ in the middle of the geodesic $\Gamma$ going from $X_1$ to $X_2$. We also fixed $d = 0.1$. We can clearly see that for this geodesic $\Gamma$, the inequality 9 is not satisfied. Therefore, $f_{Y,d}$ is not g-convex.

## C  Derivation of the gradients

### C.1  The gradient of the Riemannian distance

We start by expressing the *gradient of the squared Riemannian distance* given at Eq. 2 (see (Boumal, 2023) for the definition of a Riemannian gradient).

**Proposition C.1.** *Let $Y \in \mathcal{P}_c$ be fixed and let us consider the following mapping: $H \colon X \in \mathcal{P}_c \mapsto \delta(X, Y)^2$. Then, the* Riemannian gradient *of $H$ is*

$$\nabla H(X) = -2 \operatorname{Log}_X(Y).$$

### C.2  The gradient of Riemannian MDS

Here, we compute the gradient of the Riemannian MDS stress function. We recall that the stress function $S$ is defined as follows (see Eq. 3):

$$S(Y_1, ..., Y_n) = \sum_{i<j} (\delta(Y_i, Y_j) - D_{ij})^2. \tag{10}$$

So, using the gradient of the Riemannian distance as given in Proposition C.1, we have the following gradient:

$$\nabla_{Y_i} S(Y_1, ..., Y_N) = 2 \sum_{j \neq i} \left( \frac{D_{ij}}{\delta(Y_i, Y_j)} - 1 \right) \operatorname{Log}_{Y_i}(Y_j)$$

where we recall that $\operatorname{Log}_{Y_i}$ is the Riemannian logarithm as given in Proposition A.1.

*Proof.* We will compute the gradient with respect to $Y_k$. First, we have:

$$S(Y_1, ..., Y_n) = \sum_{i \neq k} \underbrace{(\delta(Y_i, Y_k) - D_{ik})^2}_{g_i(Y_k)} + \text{ terms independent of } Y_k.$$

Moreover, we have that the Riemannian gradient of $h_i(Y) = \delta(Y_i, Y)$ is, using C.1:

$$\nabla h_i(Y) = -\frac{\operatorname{Log}_Y(Y_i)}{\delta(Y_i, Y)}. \tag{11}$$

Therefore,

$$\nabla_{Y_k} g_i(Y_k) = 2 \left( \frac{D_{ik}}{\delta(Y_i, Y_k)} - 1 \right) \operatorname{Log}_{Y_k}(Y_i)$$

Thus, we have the full gradient:

$$\nabla_{Y_k} S(Y_1, ..., Y_N) = 2 \sum_{i \neq k} \left( \frac{D_{ik}}{\delta(Y_i, Y_k)} - 1 \right) \operatorname{Log}_{Y_k}(Y_i)$$

$\square$

We have checked numerically this gradient using the tools in PyManopt (Townsend et al., 2016). The function `check_gradient` compares the gradient to the first order Taylor expansion of the function. More details in Boumal (2023).

### C.3 Derivation of the gradient of Riemannian t-SNE

Let us now compute the gradient of the Riemannian t-SNE. We recall the following notations:

$$p_{ij} = \frac{p_{i|j} + p_{j|i}}{2} \quad q_{ij} = \frac{(1 + \delta(Y_i, Y_j)^2)^{-1}}{\sum_{k \neq l}(1 + \delta(Y_k, Y_l)^2)^{-1}}.$$

We want the gradient with respect to $Y_i$ of the following cost function:

$$C = \sum_{i=1}^{N} \text{KL}(P_i || Q_i) = \sum_{i=1}^{N} \sum_{j \neq i} p_{ij} \log \frac{p_{ij}}{q_{ij}}.$$

This gradient is given by:

$$\nabla_{Y_i} C = -4 \sum_{j=1}^{N} \frac{p_{ij} - q_{ij}}{1 + \delta(Y_i, Y_j)^2} \text{Log}_{Y_i}(Y_j).$$

Let us define the following variables to facilitate the computations:

$$d_{ij} = \delta(Y_i, Y_j)$$
$$Z = \sum_{k \neq l}(1 + \delta(Y_k, Y_l)^2)^{-1}$$

Therefore, we have $q_{ij} = \frac{(1 + d_{ij}^2)^{-1}}{Z}$. To compute the gradient $\nabla_{Y_i} C$, we use the chain rule and the symmetry of $d_{ij} = d_{ji}$:

$$\nabla_{Y_i} C = 2 \sum_{j} \frac{\partial C}{\partial d_{ij}} \nabla_{Y_i} d_{ij}. \tag{12}$$

Using the computation of the gradient of the Riemannian MDS and especially the Eq. 11, we have

$$\nabla_{Y_i} d_{ij} = -\frac{\text{Log}_{Y_i}(Y_j)}{d_{ij}}.$$

Therefore, we will focus on the computation of $\frac{\partial C}{\partial d_{ij}}$. Using the notations defined above, we have

$$C = \sum_{i=1}^{N} \sum_{j \neq i} p_{ij} \log p_{ij} - \sum_{i=1}^{N} \sum_{j \neq i} p_{ij} \log \left( \frac{(1 + d_{ij}^2)^{-1}}{Z} \right).$$

The first part of this sum does not depend on $d_{ij}$, thus, we have

$$\begin{aligned}
\frac{\partial C}{\partial d_{ij}} &= -\sum_{k=1}^{N} \sum_{l \neq k} p_{kl} \frac{\partial}{\partial d_{ij}} \log \left( \frac{(1 + d_{kl}^2)^{-1}}{Z} \right) \\
&= -\sum_{k=1}^{N} \sum_{l \neq k} p_{kl} \frac{\partial}{\partial d_{ij}} \left( -\log Z - \log(1 + d_{kl}^2) \right) \\
&= \sum_{k=1}^{N} \sum_{l \neq k} p_{kl} \frac{\partial}{\partial d_{ij}} \log Z + \sum_{k=1}^{N} \sum_{l \neq k} p_{kl} \frac{\partial}{\partial d_{ij}} \log(1 + d_{kl}^2)
\end{aligned} \tag{13}$$

Let us compute those two terms separately:

- For the first term, let us start by computing $\frac{\partial}{\partial d_{ij}} \log Z$:

$$
\begin{aligned}
\frac{\partial}{\partial d_{ij}} \log Z &= \frac{1}{Z} \frac{\partial}{\partial d_{ij}} Z \\
&= \frac{1}{Z} \frac{\partial}{\partial d_{ij}} \sum_{k \neq l} (1 + \delta(Y_k, Y_l)^2)^{-1} \quad \text{by definition of } Z \\
&= -\frac{1}{Z} \frac{2d_{ij}}{(1 + d_{ij}^2)^2} \quad \text{as the only non-zero term of the derivative of the sum is when } k=i \text{ and } l=j \\
&= -2 \frac{q_{ij}}{(1 + d_{ij}^2)} d_{ij} \quad \text{using the fact that } q_{ij} = \frac{(1+d_{ij}^2)^{-1}}{Z}.
\end{aligned}
$$

Therefore,

$$
\sum_{k=1}^{N} \sum_{l \neq k} p_{kl} \frac{\partial}{\partial d_{ij}} \log Z = -2q_{ij}(1 + d_{ij}^2)^{-1} d_{ij} \underbrace{\sum_{k=1}^{N} \sum_{l \neq k} p_{kl}}_{=1} = -2q_{ij}(1 + d_{ij}^2)^{-1} d_{ij}.
$$

- For the second term, we start by computing $\frac{\partial}{\partial d_{ij}} \log(1 + d_{kl}^2)$:

$$
\frac{\partial}{\partial d_{ij}} \log(1 + d_{kl}^2) = \frac{1}{(1 + d_{kl}^2)} \frac{\partial}{\partial d_{ij}} (1 + d_{kl}^2) = \begin{cases} 2d_{ij}(1 + d_{ij}^2)^{-1} & \text{if } k = i \text{ and } l = j \\ 0 & \text{otherwise} \end{cases}
$$

Thus,

$$
\sum_{k=1}^{N} \sum_{l \neq k} p_{kl} \frac{\partial}{\partial d_{ij}} \log(1 + d_{kl}^2) = 2d_{ij}(1 + d_{ij}^2)^{-1} p_{ij}.
$$

Stitching all together in Eq. 13, we have

$$
\frac{\partial C}{\partial d_{ij}} = -2q_{ij}(1 + d_{ij}^2)^{-1} d_{ij} + 2d_{ij}(1 + d_{ij}^2)^{-1} p_{ij} = 2d_{ij}(p_{ij} - q_{ij})(1 + d_{ij}^2)^{-1}.
$$

We can now go back to Eq 12, and we have

$$
\nabla_{Y_i} C = 2 \sum_j 2d_{ij}(p_{ij} - q_{ij})(1 + d_{ij}^2)^{-1} \left( -\frac{\mathrm{Log}_{Y_i}(Y_j)}{d_{ij}} \right) = -4 \sum_j (p_{ij} - q_{ij})(1 + d_{ij}^2)^{-1} \mathrm{Log}_{Y_i}(Y_j).
$$

As with the gradient of the MDS, we numerically checked this gradient using the `check_gradient` function of PyManopt.

## D  Proof of the affine invariance property

*Proof.* Let $(X_1, ... X_N)$ be points in $\mathcal{P}_c$ and $R$ be an invertible matrix.

1. Let $(\tilde{X}_1, ..., \tilde{X}_N) = (RX_1R^\top, ..., RX_NR^\top)$. We introduce the two following functions:

$$
S(Z_1, ..., Z_N) = \sum_{i<j} (\delta(Z_i, Z_j) - D_{ij})^2
$$

$$
\tilde{S}_R(Z_1, ..., Z_N) = \sum_{i<j} (\delta(Z_i, Z_j) - \tilde{D}_{ij})^2
$$

where $D_{ij} = \delta(X_i, X_j)$ and $\tilde{D}_{ij} = \delta(\tilde{X}_i, \tilde{X}_j)$. Then, $S$ (resp. $\tilde{S}_R$) is the function one needs to minimize when wanting to solve the Riemannian MDS problem when the initial high-dimensional points are $(X_1, ... X_{,N})$ (resp. $(\tilde{X}_1, ..., \tilde{X}_N)$). Since the Riemannian distance is affine invariant (prop. 3.3), we have

$$\tilde{D}_{ij} = \delta(\tilde{X}_i, \tilde{X}_j) = \delta(RX_iR^\top, RX_jR^\top) = \delta(X_i, X_j) = D_{ij}.$$

Therefore, $S(Z_1, ..., Z_N) = \tilde{S}_R(Z_1, ..., Z_N)$ for all $N$-tuple of SPD matrices $(Z_1, ..., Z_N)$. So the set of solutions are the same.

2. Let $(Y_1, ..., Y_N)$ be a solution of the Riemannian MDS when the initial high-dimensional points are $(X_1, ..., X_N)$. Let us denote $(\tilde{Y}_1, ..., \tilde{Y}_N) = (RY_1R^\top, ..., RY_NR^\top)$. Then, we have for all $i < j$, $\delta(Y_i, Y_j) = \delta(\tilde{Y}_i, \tilde{Y}_j)$. So, with the same notations as in point 1. We have that

$$S(Y_1, ... Y_N) = S(\tilde{Y}_1, ..., \tilde{Y}_N).$$

As $(Y_1, ..., Y_N)$ is a solution of our problem, $(\tilde{Y}_1, ..., \tilde{Y}_N)$ is too.

$\square$

# E   Reducing a geodesic

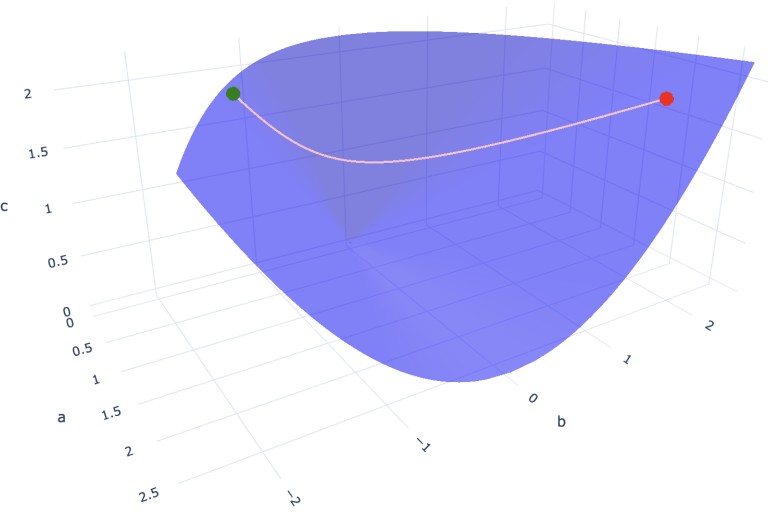

Figure 8: An example of two matrices sampled on opposite sides of the cone of SPD matrices. The pink line represents the geodesic linking those two points. They all lie in the $\mathcal{P}_c$ cone represented in blue.

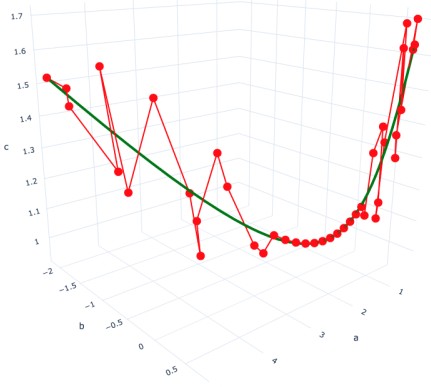

Figure 9: Results from t-SNE.

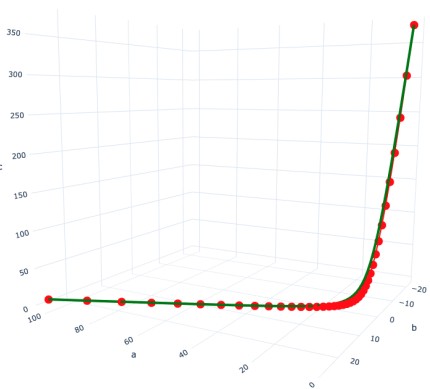

Figure 10: Results from MDS.

Figure 11: Examples of a reduced geodesic using both algorithms and their comparison with the geodesic $\tilde{\Gamma}$ going from the first to the last reduced point. The dimension of the high-dimensional space from which this example geodesic was sampled is $c = 5$. This is why, unlike in Figure 3, in this example the MDS seams to lead to better results than the t-SNE.

In experiment, we investigate how a high-dimensional geodesic is transformed using our algorithms. To be more precise, we sample a random geodesic $\Gamma$ in $\mathcal{P}_c$ going from two randomly sampled points $X_1$ and $X_N$, the start and end point of the geodesic. We want $X_1$ and $X_N$ to be on "opposite sides" of the cone of SPD matrices. To do that, we will use their spectral decomposition. We start by sampling $U$ uniformly in $O(d)$ (Mezzadri, 2007). We then sample the eigen values $(\lambda_1^1, ...\lambda_d^1)$ (resp. $(\lambda_1^N, ...\lambda_d^N)$) of $X_1$ (resp. $X_N$). To do that, we sample the first half of the eigen values of $X_1$ in $\mathcal{U}([3, 4])$ and the second half in $\mathcal{U}([0, 1])$. For $X_N$, we do the opposite, the first half of the eigen values are drawn from $\mathcal{U}([0, 1])$ and the second half from $\mathcal{U}([3, 4])$. We can then compute $X_1$ (resp. $X_N$) as $X_1 = U\mathrm{diag}(\lambda_1^1, ..., \lambda_d^1)U^\top$ (resp. $X_N = U\mathrm{diag}(\lambda_1^N, ..., \lambda_d^N)U^\top$). We give an example of matrices sampled on "opposite sides" of the cone in Figure 8.

Once the initial and final point $X_1$ and $X_N$ of the geodesic are sampled, we consider $(X_1, ..., X_N)$, $N$ points uniformly spaced on the geodesic $\Gamma$. We reduce these points using our algorithms into $(Y_1, ...Y_N)$, $N$ points in $\mathcal{P}_2$. We compare the trajectory $(Y_1, ...Y_N)$ to the geodesic $\tilde{\Gamma}$ in $\mathcal{P}_2$ going from $Y_1$ to $Y_N$. To do this, we consider $(\tilde{Y}_1, ..., \tilde{Y}_N)$ uniformly spaced on the geodesic $\tilde{\Gamma}$ in $\mathcal{P}_2$ (therefore $\tilde{Y}_i = \tilde{\Gamma}(\frac{i-1}{N-1})$). Then, we compute the distances between the points $(Y_1, ...Y_N)$ and $(\tilde{Y}_1, ..., \tilde{Y}_N)$ that we denote $d_{\text{point}}((Y_i)_i, \tilde{\Gamma})$:

$$d_{\text{point}}((Y_i)_i, \tilde{\Gamma}) = \sum_{i=1}^{N} \delta(Y_i, \tilde{Y}_i)^2 \tag{14}$$

Given this setup, we conducted some experiments varying some parameters: the dimension $c$ of the true geodesic, the noise $\varepsilon$ on the sampling of the initial points on the true geodesic and the number of points $N$ sampled on the geodesic. The base parameters are $c = 50$, $\varepsilon = 0$ (no noise) and $N = 100$. The perplexity of the Riemannian t-SNE was set to be $\frac{3}{4}N$ for all the experiments (see Appendix F).

## F   Influence of the perplexity

In this section we discuss the perplexity parameter that plays an important role in the t-SNE algorithm. We recall that the perplexity is a hyperparameter chosen by the user and that will be used to compute the variance of the Gaussians in the conditional probabilities defined in Eq. 4. The role of the perplexity is to fix the entropy of the conditional probabilities. As explained in (Van Assel et al., 2023), the perplexity can be interpreted as the effective number of neighbors for each data point. It is a way of balancing attention between some local and some global aspects of the data one wants to reduce. It is known that varying this parameter can change a lot the results one gets from the t-SNE, and there is no silver bullet method to choose the correct perplexity (Wattenberg et al., 2016).

In the Euclidean case, van der Maaten & Hinton (2008) say that "The performance of SNE is fairly robust to changes in the perplexity, and typical values are between 5 and 50". We want to see if this statement is still true in the Riemannian case. For this, we sample two Riemannian Gaussian in $\mathcal{P}_c$ where $c = 5$ is the high-dimensional space. The first Gaussian has for mean $\bar{X}_1 = I_c$ (the identity matrix) and a variance of $\sigma_1 = 0.5$ and the second one has a mean of $\bar{X}_1 = 10I_c$ and a variance of $\sigma_1 = 0.25$. We sample $N = 60$ points in each Gaussian and concatenate them together to create the high-dimensional dataset. We then use the Riemannian version of the t-SNE algorithm with different values of the perplexity $\texttt{perp} \in [5, 10, 15, ..., 50, 55]$. The initial point of the optimization process is fixed to only see the influence of the perplexity. We show the results in Figure 12. We see that the perplexity has an important impact on the result. When the perplexity is too small, the points the further from the origin have huge coefficients and are all aligned. As the perplexity grows, the coefficients of the points get smaller and smaller, and we are able to see some structure in the dataset appear. Contrary to the Euclidean t-SNE, we need a larger perplexity to have a convincing result. However, when the perplexity is too big, here where it is equal to 120, all the point cloud gets flatten into a hyperplane, all the points are then aligned in a 2D plan. Throughout our experiments, we found empirically that taking the perplexity to be equal to 3/4 of the total number of points works well. In this example, it corresponds to taking a perplexity of 90, which seems to correctly reproduce the desired structure.

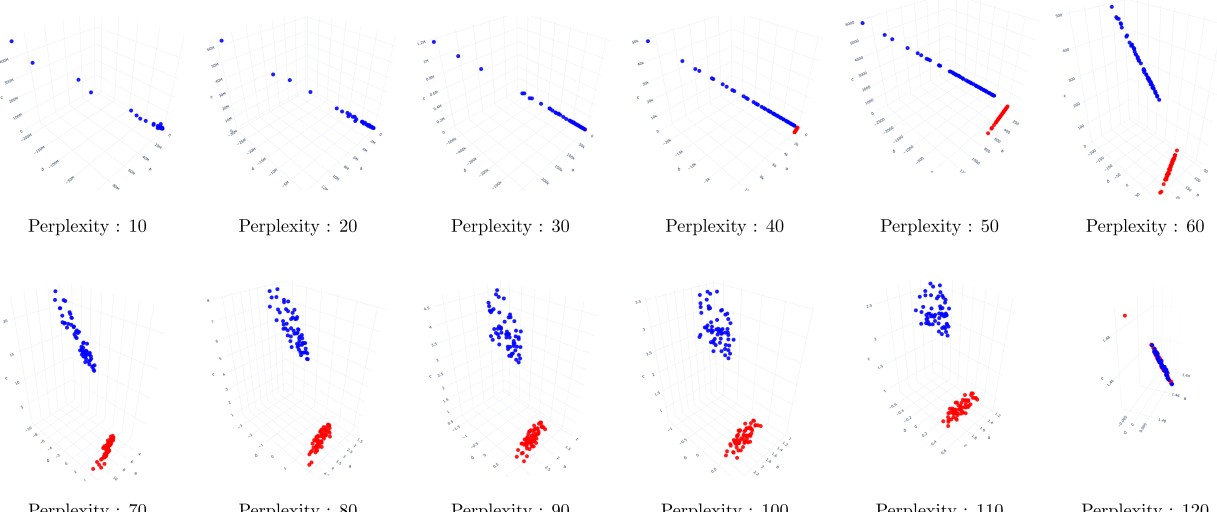

| Perplexity : 10 | Perplexity : 20 | Perplexity : 30 | Perplexity : 40 | Perplexity : 50 | Perplexity : 60 |

| Perplexity : 70 | Perplexity : 80 | Perplexity : 90 | Perplexity : 100 | Perplexity : 110 | Perplexity : 120 |

Figure 12: We show in this plot the same high-dimensional point cloud $X$ that is reduced using the Riemannian t-SNE algorithm with different choices of the perplexity. The colors represent the two Gaussians from which we sampled $X$. On the first three plots, the red points are not visible, it is because the blue points have very big coefficients, and the red ones small coefficients, so we are not able to see both of them at the same time.

## G  Details on the experiments on real datasets

In this section, we give details on the different datasets we used in the experiments of section 7.4. Then, we give some information on the computation times of the different algorithms.

### G.1  Details on the datasets

We give here a brief introduction to each of the 7 datasets used in our experiments at section 7.4.

**BCI datasets:**  3 of the datasets are coming from Brain Computer Interfaces (BCI) experiments: BNCI2014001 (Tangermann et al., 2012), BNCI2014002 (Steyrl et al., 2015) and AlexMI (Barachant, 2012). These datasets come from MOABB (Aristimunha et al., 2023). The selected 3 motor imagery datasets consist of several subjects for each dataset and several sessions for each subject. The two classes of each dataset are balanced and the size of the matrices are respectively $22 \times 22$, $15 \times 15$ and $16 \times 16$. We start by applying a standard band-pass filter with range $[7; 35]$ Hz for each dataset. To compute the covariance matrices from an EEG, we used the Ledoit-Wolf shrunk covariance matrix (Ledoit & Wolf, 2004) to avoid ill-conditioned matrices.

**AirQuality dataset:**  This dataset is from the Beijing Municipal Monitoring Center. It is a dataset of air quality monitored from 34 different sites in Beijing, China (Hua et al., 2021). For each site, six atmospheric pollutants where recorded every hour: CO, $NO_2$, $O_3$, $PM_{10}$, $PM_{2.5}$ and $SO_2$. We used the same preprocessing as in (Smith et al., 2022a) to get a point cloud of 102 covariance matrices of size $6 \times 6$. Each covariance matrix has a label depending on which period it represents: weekdays, weekends or holidays.

**FPHA dataset:**  This is a dataset of video sequences in first person view used as benchmark for estimating hand actions. The original dataset (Garcia-Hernando et al., 2018) consist of 1,175 video sequences distributed across 45 distinct categories. We chose 8 of those categories for a total of 108 videos. The categories chosen are `drink_mug`, `give_card`, `give_coin`, `high_five`, `open_juice_bottle`, `open_peanut_butter`, `put_sugar`, `write_pen`. As in (Wang et al., 2023), we converted the video sequences into $63 \times 63$ SPD matrices representing how the 3D coordinates of 21 hand joints are linked together.

**TEP dataset:** The last dataset is obtained from a simulation of an industrial process known as the Tennessee Eastman Process (TEP) (Downs & Vogel, 1993). This dataset is used as a benchmark dataset testing and comparing various anomaly detection methods. We used the preprocessing described in Smith et al. (2022b) to get a dataset containing 420 SPD matrices of size $52 \times 52$.

**The setup**  Once the datasets are reduced using the different algorithms, we want to compute the trustworthiness coefficient. For this, we need to choose a hyperparameter $k$ for the number of nearest neighbors. As the different dataset have different numbers of points $N$, in order to compare them we choose $k$ to be a certain percentage of the total number of points $N$. We choose for our study $k \in \{0.05N, 0.1N, 0.2N, 0.3N, 0.4N, 0.5N\}$ as one needs $k \leq N/2$. All the results are given at Appendix H.

## H   Detailed table of the results of the different algorithms on the different datasets.

| Dataset | Algorithm | Trustworthiness with the neighborhood as a % of the total number of points | | | | | |
| | | 0.05 | 0.1 | 0.2 | 0.3 | 0.4 | 0.5 |
|---|---|---|---|---|---|---|---|
| AirQuality | Riem-Riem t-SNE | 0.9751 | 0.9822 | 0.9890 | 0.9924 | 0.9894 | 0.9899 |
| | Euc-Euc t-SNE | 0.8785 | 0.8776 | 0.8819 | 0.8902 | 0.8771 | 0.8519 |
| | Riem-Euc t-SNE | **0.9814** | 0.9834 | 0.9895 | 0.9895 | 0.9800 | 0.9739 |
| | Riem-Riem MDS | 0.9783 | 0.9836 | 0.9919 | 0.9950 | **0.9948** | 0.9935 |
| | Euc-Euc MDS | 0.8747 | 0.8737 | 0.8772 | 0.8835 | 0.8657 | 0.8332 |
| | Riem-Euc MDS | 0.9811 | **0.9834** | **0.9914** | 0.9933 | 0.9945 | 0.9932 |
| | UMAP | 0.9549 | 0.9580 | 0.9627 | 0.9567 | 0.9395 | 0.9439 |
| | PGA | 0.9665 | 0.9735 | 0.9885 | **0.9949** | 0.9946 | **0.9945** |
| | Riem PCA | 0.9381 | 0.9503 | 0.9683 | 0.9750 | 0.9745 | 0.9706 |
| AlexMI | Riem-Riem t-SNE | 0.9479 | 0.9439 | **0.9461** | **0.9435** | **0.9404** | 0.9311 |
| | Euc-Euc t-SNE | 0.7622 | 0.7551 | 0.7592 | 0.7620 | 0.7549 | 0.7351 |
| | Riem-Euc t-SNE | **0.9653** | **0.9530** | 0.9366 | 0.9289 | 0.9210 | 0.9052 |
| | Riem-Riem MDS | 0.9127 | 0.9072 | 0.9087 | 0.9092 | 0.9064 | 0.8946 |
| | Euc-Euc MDS | 0.7585 | 0.7577 | 0.7668 | 0.7747 | 0.7754 | 0.7557 |
| | Riem-Euc MDS | 0.9166 | 0.9143 | 0.9133 | 0.9127 | 0.9100 | 0.8965 |
| | UMAP | 0.9252 | 0.9123 | 0.8930 | 0.8751 | 0.8577 | 0.8286 |
| | PGA | 0.8851 | 0.9018 | 0.9204 | 0.9299 | 0.9341 | **0.9332** |
| | Riem PCA | 0.7804 | 0.7924 | 0.8189 | 0.8316 | 0.8381 | 0.8268 |
| BNCI2014001 | Riem-Riem t-SNE | 0.9488 | **0.9543** | **0.9605** | **0.9623** | **0.9599** | 0.9419 |
| | Euc-Euc t-SNE | 0.7075 | 0.6937 | 0.6938 | 0.6972 | 0.6961 | 0.6684 |
| | Riem-Euc t-SNE | **0.9564** | 0.9454 | 0.9351 | 0.9219 | 0.9036 | 0.8376 |
| | Riem-Riem MDS | 0.9303 | 0.9333 | 0.9367 | 0.9366 | 0.9327 | 0.9205 |
| | Euc-Euc MDS | 0.7048 | 0.7071 | 0.7160 | 0.7203 | 0.7175 | 0.6891 |
| | Riem-Euc MDS | 0.9304 | 0.9334 | 0.9360 | 0.9355 | 0.9310 | 0.9182 |
| | UMAP | 0.9408 | 0.9316 | 0.9194 | 0.9035 | 0.8782 | 0.8062 |
| | PGA | 0.9231 | 0.9319 | 0.9442 | 0.9511 | 0.9528 | **0.9448** |
| | Riem PCA | 0.7762 | 0.7825 | 0.7945 | 0.8011 | 0.8008 | 0.7815 |
| BNCI2014002 | Riem-Riem t-SNE | 0.9583 | **0.9629** | **0.9665** | **0.9676** | **0.9648** | 0.9549 |
| | Euc-Euc t-SNE | 0.7370 | 0.7239 | 0.7191 | 0.7155 | 0.7067 | 0.6663 |
| | Riem-Euc t-SNE | **0.9672** | 0.9553 | 0.9451 | 0.9335 | 0.9177 | 0.8808 |
| | Riem-Riem MDS | 0.9376 | 0.9398 | 0.9429 | 0.9426 | 0.9386 | 0.9288 |
| | Euc-Euc MDS | 0.7429 | 0.7428 | 0.7437 | 0.7434 | 0.7346 | 0.6992 |
| | Riem-Euc MDS | 0.9375 | 0.9406 | 0.9438 | 0.9434 | 0.9391 | 0.9279 |
| | UMAP | 0.9476 | 0.9358 | 0.9166 | 0.8964 | 0.8731 | 0.8260 |
| | PGA | 0.9317 | 0.9403 | 0.9527 | 0.9590 | 0.9610 | **0.9575** |
| | Riem PCA | 0.8338 | 0.8455 | 0.8623 | 0.8729 | 0.8735 | 0.8590 |
| FPHA | Riem-Riem t-SNE | 0.9652 | 0.9685 | **0.9704** | **0.9763** | **0.9771** | 0.9707 |
| | Euc-Euc t-SNE | 0.9114 | 0.8771 | 0.8205 | 0.7711 | 0.7143 | 0.6034 |
| | Riem-Euc t-SNE | **0.9845** | **0.9722** | 0.9645 | 0.9616 | 0.9587 | 0.9348 |
| | Riem-Riem MDS | 0.8955 | 0.8889 | 0.9033 | 0.9269 | 0.9464 | 0.9610 |
| | Euc-Euc MDS | 0.8665 | 0.8477 | 0.7935 | 0.7345 | 0.6906 | 0.6023 |
| | Riem-Euc MDS | 0.9525 | 0.9500 | 0.9462 | 0.9439 | 0.9441 | 0.9395 |
| | UMAP | 0.9695 | 0.9581 | 0.9486 | 0.9369 | 0.9268 | 0.8989 |
| | PGA | 0.9512 | 0.9551 | 0.9626 | 0.9686 | 0.9729 | **0.9766** |
| | Riem PCA | 0.8490 | 0.8247 | 0.8174 | 0.8151 | 0.8080 | 0.7843 |
| TEP | Riem-Riem t-SNE | 0.9842 | 0.9804 | 0.9780 | 0.9804 | 0.9761 | 0.9750 |
| | Euc-Euc t-SNE | 0.8892 | 0.8849 | 0.9043 | 0.9264 | 0.9379 | 0.9465 |
| | Riem-Euc t-SNE | 0.9938 | 0.9833 | 0.9797 | 0.9686 | 0.9328 | 0.9242 |
| | Riem-Riem MDS | 0.9732 | 0.9621 | 0.9755 | 0.9844 | 0.9901 | 0.9949 |
| | Euc-Euc MDS | 0.8902 | 0.8931 | 0.9061 | 0.9146 | 0.9186 | 0.9075 |
| | Riem-Euc MDS | **0.9945** | **0.9916** | **0.9921** | **0.9930** | **0.9927** | 0.9890 |
| | UMAP | 0.9788 | 0.8448 | 0.7128 | 0.6287 | 0.5878 | 0.4989 |
| | PGA | 0.9743 | 0.9736 | 0.9767 | 0.9781 | 0.9849 | **0.9909** |
| | Riem PCA | 0.9143 | 0.8998 | 0.9143 | 0.9355 | 0.9499 | 0.9575 |

# I    Plot of the results of the dataset AirQuality

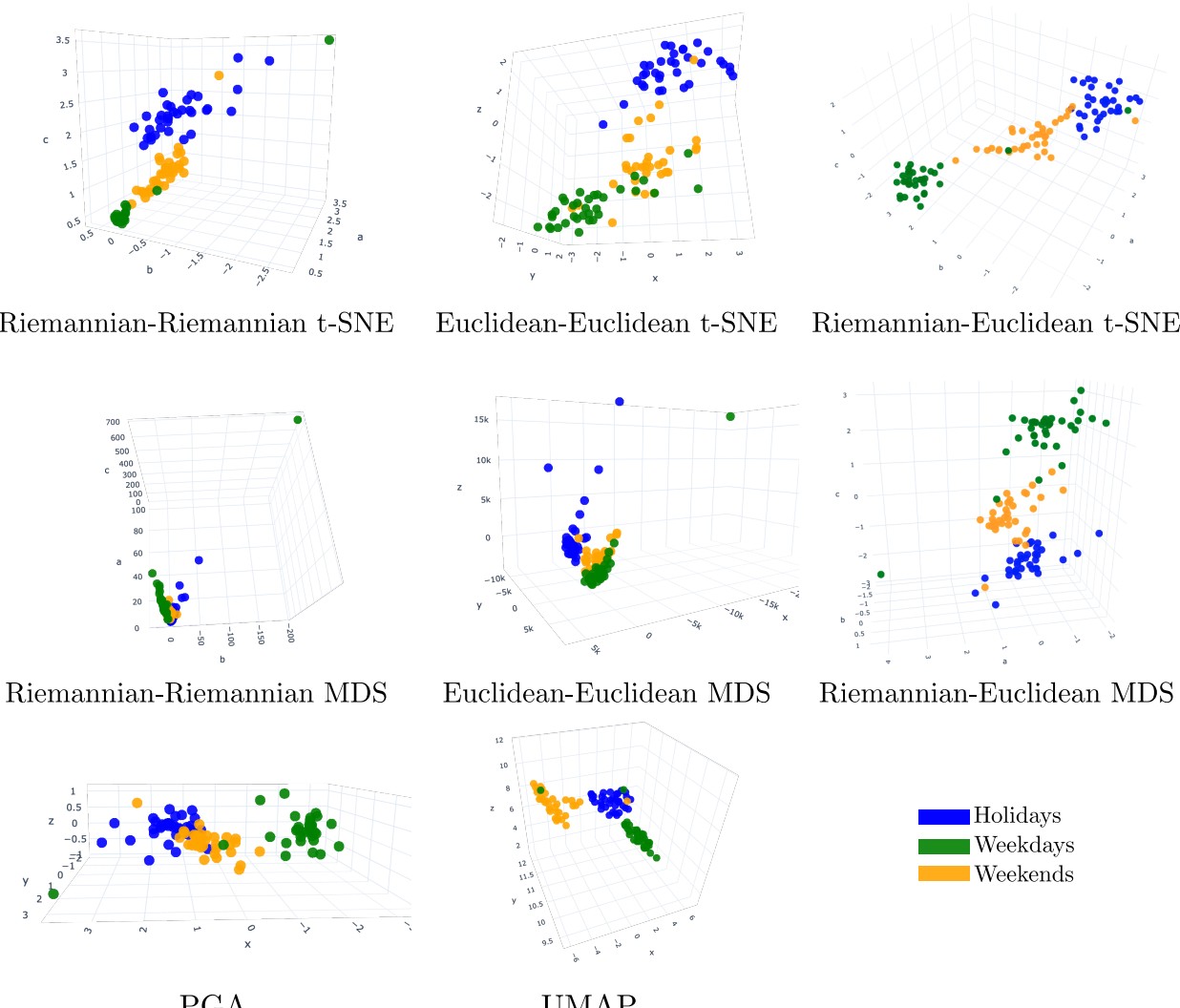

Figure 13: Reduced point cloud of the dataset AirQuality using the different algorithms.

