# OpenReview forum: "Geometry-Aware visualization of high dimensional Symmetric Positive Definite matrices"
_TMLR — Accepted by TMLR_

### Review · Reviewer_asDR · 2024-10-22

**Summary Of Contributions:**

In this work, the authors address a significant limitation of the commonly used visualization techniques, t-SNE and Multidimensional Scaling (MDS), when applied to Symmetric Positive Definite matrices (SPD). Specifically, they observe that due to the curvature of the Riemannian manifold of SPD matrices, t-SNE and MDS distort the relative distances of the visualized points since they project them into a flat Euclidean space. To solve this problem they propose a novel extension to the t-SNE and Multidimensional Scaling (MDS) algorithms that respects the Riemannian structure of the manifold where the SPD matrices live. In addition to a detailed description of the proposed extension, the authors provide theoretical results regarding the invariance properties of the proposed algorithms and their convergence. Finally, the authors offer qualitative and quantitative experimental results showcasing the benefits of using the proposed algorithms compared to their Euclidean counterparts.

**Audience:**

Yes

**Broader Impact Concerns:**

There are no concerns on the ethical implications of the work that would require adding a Broader Impact Statement.

**Claims And Evidence:**

Yes

**Requested Changes:**

Some recommended changes of medium importance:
- A more detailed discussion of the time and memory resources required by the proposed algorithm compared to the Euclidean counterparts will add to the completeness of the presentation of the proposed algorithms.
- Regarding the second weakness: it would be valuable to provide a qualitative comparison of the visualization using the full Riemannian versions of the algorithms and the Riemannian-Euclidean versions.

**Strengths And Weaknesses:**

Strengths
- The authors provide a comprehensive description of the currently commonly used visualization methods and showcase how they can result in distortion of the underlying data structure for the case of SPD matrices.
- The presentation of the proposed algorithms is clear. Additionally, the investigation of their convergence and invariance properties adds to the completeness of this work.
- The experimental results provide significant evidence regarding the benefits of the proposed extension of t-SNE and MDS for SPD matrices.

Weaknesses
- There is limited discussion and comparison of the convergence speed and computational resources of the proposed algorithms compared to their Euclidean counterparts. Are there any computational tradeoffs for the proposed methods or they are the same as the original visualization algorithms?
- In the qualitative results of Figure 2, only the full Riemannian version is shown and compared with the Euclidean version. Since in the full Riemannian version the distance between the projected points is given by the geodesic presented in Figure 1b, it might be still difficult for a user to inspect the relative distances between these points since the geodesic distance cannot be intuitively inferred by visual inspection. Is it possible that the Riemannian-Euclidean version is a better tradeoff for visualization purposes although it might be worse in the trustworthiness metric?

---

> ### Author Response · Authors · 2024-11-04
>
> - Regarding the convergence speed and computational resources of the proposed algorithms, the Riemannian version requires more time to converge as each iteration is longer to compute. Indeed, at each iteration of the gradient descent, the matrix distance needs to be computed between all the points. As the AIRM distance requires to compute matrix inverse square root and matrix logarithm, it takes more time to compute than its Euclidean counterpart. However, one should remark that the dimension of the high-dimensional points has very little influence on the time each iteration takes to computes indeed, at each iteration, only 2 x 2 SPD matrices are at stake, so the distance matrix is always computed on 2 x 2 matrices. We also remarked that the Riemannian t-SNE needed a lot less iterations to convergence than the Riemannian MDS.
> Additionally, it is important to highlight that Euclidean MDS has a closed-form solution based on the eigendecomposition of the distance matrix, leveraging the Euclidean properties of the distance measure. In contrast, Riemannian MDS lacks a closed-form solution, so solving the optimization problem requires a Riemannian gradient descent, as implemented here.
> We can provide more quantitative data on the computation time and number of iterations needed for each algorithm to converge on the different datasets.
>
> - Concerning the second remark, we will provide more plots to fully visualize the difference between the full Riemannian versions of the algorithms and the Riemannian-Euclidean versions.

---

### Review · Reviewer_bzCj · 2024-10-25

**Summary Of Contributions:**

The authors formulate two dimensionality reduction algorithms, Riemannian Multidimensional Scaling, and Riemannian t-SNE. They provide a proof that both algorithms are invariant to affine transformations, which is a major advantage when comparing data across sensor array instances that vary spatially. Both algorithms are solved via Riemanian gradient descent, and for Riemannian MDS a discussion on convergence is provided.

**Audience:**

Yes

**Broader Impact Concerns:**

no concerns

**Claims And Evidence:**

Yes

**Requested Changes:**

# Requested Changes
* Sec 7.3 "Reducing geodesics": the observation that MDS "seems to do a better job than the t-SNE" seems at odds with the figures in that section, which shows t-SNE being more robust to noise, dimension, and N-points. I see in Figure 11 what appears to be MDS "fitting" better to the manifold but I'm having trouble reconciling this with Fig 3. A more unified conclusion (not split across main/appendix) might help clarify this result.
* You describe trustworthiness in one sentence as "It ranges within [0, 1], and measures how well the ranks of the k-nearest neighbors of each point are preserved." I think it's worth adding a little more detail since this is the key performance comparator. Is this the rank of the distance /adjacency matrix?

# Comments/Observations (not critical to recommendation)
* The paper is very clean so I understand not wanting to expand too much discussion. But comparison / contextualization with information geometry terminology may be interesting (Fisher Matrix, Natural Gradient Descent)
* Figure 4 is a very fair visualization; is there a way to somehow capture the falloff of PGA and UMAP that is shown in Fig 5?
* Any reason why you didn't try UMAP with a Riemannian geodesic-based distance?
* Does Sec 8 deserve its own section as opposed to simply adding it to the competitors above? (I appreciate the comparison, just asking)

**Strengths And Weaknesses:**

# Strengths
* Nice intro, with complete set of references.
* Clean formulations, propositions, and discussion with a variety of visualizations.
* Fair comparisons with a number of well-known competitors.

# Weaknesses
* No significant weaknesses. A couple of comments in the below Requested Changes section regarding clarity on metrics.

---

> ### Author Response · Authors · 2024-11-04
>
> ### Regarding the requested changes:
>
> - For the “Reducing geodesics” experiments, the fact that, in figure 11, the MDS looks better than the t-SNE is mainly due to the dimension of the high-dimensional space from which the geodesic has sampled: for this figure, the dimension of the SPD matrices are 5 x 5. If one looks at the left plot at figure 3, one can see that, when the dimension c=5, the Riemannian MDS lead to better results than the Riemannian t-SNE. In the final version, we will add this important information that answers any misinterpretation.
>
> - To explain in more detail how the trustworthiness coefficient works, we propose to modify the previous sentence by: It ranges from 0 to 1 (with 1 being the highest score) and is computed by ranking the order of each point’s neighbors and comparing these ranks between the original high-dimensional space and the reduced low-dimensional space. Any nearest neighbors in the reduced space that differ from the original space are penalized according to their ranking in the input space.
>
> ### Regarding the comments:
>
> - To add some contextualization with information geometry terminology, we will add that the AIRM geometry we use on the manifold of SPD matrices is the Fisher-Rao metric on Multivariate Normals [1].
> - The falloff of the PGA and UMAP can be seen in the supplementary materials, in which we provide 3D visualizations of the reduced Gaussian mixture used in the first synthetic experiment using the different algorithms (such as UMAP and PGA).
> - The dimension reduction using UMAP is done with the Riemannian distance matrix as an input. It might indeed not be clear in section 7.1, so we will clarify it in a final version.
> - For the remark on section 8, we initially added the log-euclidian versions of the algorithms to the competitors, but we felt that it added too much complexity to the plots and the comparison. By dividing it into two separate sections, we hope that our main message (one should use Riemannian aware dimension reduction algorithm when dealing with SPD matrices, whatever the metric used) will be better delivered to the reader.
>
> [1] Lene Theil Skovgaard. “A Riemannian Geometry of the Multivariate Normal Model.” Scandinavian Journal of Statistics, vol. 11, no. 4, 1984.

---

### Review · Reviewer_Zj4X · 2024-11-17

**Summary Of Contributions:**

This paper introduces Riemannian versions of t-SNE and Multidimensional Scaling for visualizing Symmetric Positive Definite (SPD) matrices while preserving their manifold curvature. The experimental results demonstrate that these methods successfully maintain the geometric properties of SPD Gaussians and geodesics on both synthetic and real datasets.

**Audience:**

Yes

**Claims And Evidence:**

Yes

**Requested Changes:**

The paper would benefit from more professional use of parenthetical in-text citations to improve readability; for instance, in the first paragraph, it should be “in process control (Willjuice Iruthayarajan & Baskar, 2010)”. Additionally, Figure 1 lacks a main caption and figure number, with only subcaptions labeled (a) and (b) currently provided.

Another concern is the performance impact of Riemannian gradient descent on high-dimensional product manifolds when N is large. The paper should address potential issues related to convergence speed and guarantees, as well as any implications for practical scalability. The current Section 6 on convergence is brief and does not sufficiently address the performance or stability of the gradient descent algorithm. The DCA algorithm is also mentioned, but it does not appear to be implemented in the experiments; clarification on this would be helpful. Additionally, while the authors state that the method will converge to a critical point, it is unclear whether this point is guaranteed to be a minimizer, which is important for both the MDS and t-SNE approaches. The discussion on computational time could also be expanded in the numerical results section, including practical details such as the number of iterations typically required.

In Figure 3, the behavior of Riemannian t-SNE seems unaffected by changes in dimension or noise level. Could the authors provide a more intuitive explanation for this stability? It seems unlikely to be solely due to the robustness of the Riemannian t-SNE method, as it appears remarkably invariant across varied conditions.

**Strengths And Weaknesses:**

Strengths: a new method for visualizing SPD while preserving their curvature as compared with Euclidean approaches. Extensive synthetic and real-data evaluations of the proposed method.

Weakness: The writing requires substantial improvement. And this work lacks sufficient detail and guarantee for the convergence results.

---

> ### Author Response · Authors · 2024-11-20
>
> - In our final version, we will make sure to better use parentheses, especially on in-text citations. We will also include a main caption to the first figure.
> - Regarding the convergence speed of our algorithms, we will, as mentioned in the comment of the review of reviewer asDR, give more quantitative data on the computation times of our algorithms on the different datasets.
> - Indeed, we mention in section 6 a convergence result that could be achieved using a Riemannian DCA. In the experiments, we did not implement this DCA, but rather a Riemannian Gradient Descent (RGD) algorithm. We do not have any theoretical guarantees on the convergence of this problem using a RGD, however, in practice, we stopped the RGD when the gradient was small enough (< 1e-06). - The goal of our convergence section was to register our problem in a class of optimization problems, that is the class of problems that can be written as a difference of convex functions. There are no guarantees either that the critical point that is reached is indeed a minimizer of our cost function.
> - Regarding the final comment on Figure 3, t-SNE is not entirely robust to noise and changes in dimensionality. However, it demonstrates significantly greater robustness compared to MDS, which is highly sensitive to these two factors. This contrast is clearly illustrated in Figure 3. The relative robustness of t-SNE to noise comes from the fundamental design of the algorithm. By employing a Student t-distribution in the low-dimensional space, t-SNE prioritizes preserving the relationships of points that are close to each other while giving much less weight to points that are farther apart. This focus on local structure inherently buffers the impact of noise, as minor perturbations in distant points have a reduced effect on the embedding. In contrast, MDS aims to preserve pairwise distances across the entire dataset equally, regardless of whether the points are near or far. As a result, MDS is more vulnerable to noise and dimensionality changes, which disproportionately affect the global structure of the data. This key difference explains the superior robustness of t-SNE relative to MDS in the scenarios depicted in Figure 3.

---

### Author Response · Authors · 2025-02-04

First, we would like to thank the reviewers for their work and relevant questions.
We uploaded a new version of the paper that takes into account the different reviews and where the changes are notified in red. Let us detail the main changes we made:
- We added in figure 2 and in figure 13 visualizations of the Riemannian-Euclidean approach as suggested by reviewer asDR. We also add a analysis of this visualizations in the text of section 7.3.
- We added section 7.5 as well as Table 2 where we give more quantitatif results on the computation times of the different algorithms as suggested by reviewer Zj4X and asDR.
- We modified all the uses of parenthetical in-text citations as suggested by reviewer Zj4X.
- We also added a contextualization with information geometry terminology in section 3 and modified the description of `trustworthiness` in section 7.2 as suggested by reviewer bzCj.
Finally, we made some minor modifications on the text to improve the readability of the paper.

---

> ### Comment · Action_Editor_hgAU · 2025-02-04
> **Green light to submit**
>
> Dear authors,
>
> Please submit the camera-ready version.
>
> Best,

---

> > ### Author Response · Authors · 2025-02-06
> >
> > Hello,
> >
> > We have just submitted the camera-ready version.
> >
> > Best,

---

### Decision · Action_Editor_hgAU · 2025-01-20

**Recommendation:** Accept with minor revision

**Comment:**

The paper introduces Riemannian variants of the MDS and t-SNE algorithms, enabling visualization of SPD matrices from higher dimensions to 2x2 matrices. Additionally, it discusses which metric is more suitable. Overall, the reviewers appreciated the paper and its contributions.

However, reviewers Zj4X and asDR have raised concerns regarding certain promised revisions that have not been addressed in the paper, even though they were acknowledged in the comments.

1. The authors committed to providing visualizations that illustrate the qualitative differences between the Euclidean, full Riemannian, and Riemannian-Euclidean approaches.
2. The paper lacks a detailed discussion on the convergence speed and theoretical insights of the implemented algorithms. Additionally, while the response was useful, it is somewhat brief and lacks concrete details such as a quantitative comparison of convergence rates and computational times.

It would be beneficial if these points could be included in the revised draft version.

**Audience:**

Yes, it is appropriate.

**Claims And Evidence:**

Yes, the claims are validated.